

# Overview of experiment design and comparison of models participating in phase 1 of the SPARC Quasi-Biennial Oscillation initiative (QBOi)

Neal Butchart[1], James A. Anstey[2], Kevin Hamilton[3], Scott Osprey[4], Charles McLandress[5,2], Andrew C. Bushell[6], Yoshio Kawatani[7], Young-Ha Kim[8], Francois Lott[9], John Scinocca[2], Tim Stockdale[10], Omar Bellprat[11], Peter Braesicke[12], Chiara Cagnazzo[13], Chih-Chieh Chen[14], Hye-Yeong Chun[15], Mikhail Dobrynin[16], Rolando R. Garcia[14], Javier Garcia-Serrano[11], Lesley J. Gray[4], Laura Holt[17], Tobias Kerzenmacher[12], Hiroaki Naoe[18], Holger Pohlmann[19], Jadwiga H. Richter[14], Adam A. Scaife[1,20], Verena Schenzinger[4], Federico Serva[13,21], Stefan Versick[12], Shingo Watanabe[7], Kohei Yoshida[18], and Seiji Yukimoto[18]

[1]Met Office Hadley Centre (MOHC), Exeter, UK
[2]Canadian Centre for Climate Modelling and Analysis (CCCma), Victoria, Canada
[3]International Pacific Research Center (IPRC), Honolulu, USA
[4]National Centre for Atmospheric Science (NCAS), University of Oxford, Oxford, UK
[5]University of Toronto, Toronto, Canada
[6]Met Office, Exeter, UK
[7]Japan Agency for Marine-Earth Science and Technology (JAMSTEC), Yokohama, Japan
[8]Ewha Womans University, Seoul, South Korea
[9]Laboratoire de Météorologie Dynamique (LMD), Paris, France
[10]European Centre for Medium-Range Weather Forecasts (ECMWF), Reading, UK
[11]Barcelona Supercomputing Center (BSC), Barcelona, Spain
[12]Karlsruher Institut für Technologie (KIT), Karlsruhe, Germany
[13]Istituto di Scienze Dell'Atmosfera e del Clima (ISAC), Roma, Italy
[14]National Center for Atmospheric Research (NCAR), Boulder, USA
[15]Yonsei University, Seoul, South Korea
[16]Universität Hamburg, Hamburg, Germany
[17]NorthWest Research Associates (NWRA), Boulder, USA
[18]Meteorological Research Institute (MRI), Tsukuba, Japan
[19]Max-Planck-Institut für Meteorologie (MPI), Hamburg, Germany
[20]University of Exeter, Exeter, UK
[21]Università degli Studi di Napoli "Parthenope," Napoli, Italy

*Correspondence to:* Neal Butchart (neal.butchart@metoffice.gov.uk)

**Abstract.** The Stratosphere-troposphere Processes And their Role in Climate (SPARC) Quasi-Biennial Oscillation initiative (QBOi) aims to improve the fidelity of tropical stratospheric variability in general circulation and Earth system models by conducting coordinated numerical experiments and analysis. In the equatorial stratosphere, the QBO is the most conspicuous mode of variability. Five coordinated experiments have therefore been designed to (i) evaluate and compare the verisimilitude of

5   modelled QBOs under present-day conditions, (ii) identify robustness (or, alternatively the spread/uncertainty) in the simulated QBO response to commonly imposed changes in model climate forcings (e.g., a doubling of $CO_2$ amounts) and, (iii) examine



model dependence of QBO predictability. This paper documents these experiments and the recommended output diagnostics. The rationale behind the experimental design and choice of diagnostics is presented. To facilitate scientific interpretation of the results in other planned QBOi studies, consistent descriptions of the models performing each experiment set are given, with those aspects particularly relevant for simulating the QBO tabulated for easy comparison.

## 1 Introduction

Over the last decade, or so, there has been a move toward global climate, Earth-system, and weather-forecasting models having properly resolved stratospheres and elevated upper boundaries. In some cases (e.g., Marsh et al., 2013) these boundaries are above 100 km and thus nominally located in Space (as defined by the Fédération Aéronautique Internationale). Despite this, tropical stratospheric variability and in particular the Quasi-Biennial Oscillation (QBO) has generally been rather poorly represented (Butchart et al., 2011) in models used in recent international assessments of stratospheric ozone depletion (WMO, 2011, 2015). Likewise only a handful of the models central to the last international assessment of climate change (IPCC, 2013) simulated tropical variability approaching a realistic QBO (see Figure 1). Even with the latest generation of models the representation of the QBO remains problematic in many cases (Schenzinger et al., 2017). For instance, several of the state-of-the-art chemistry-climate models participating in the concurrent Chemistry-Climate Model Initiative (CCMI) prescribe a QBO in order to "improve" the accuracy of their simulations (Morgenstern et al., 2017). Consequently the World Climate Research Programme (WCRP) Stratosphere-troposphere Processes And their Role in Climate (SPARC) core project has promoted a new QBO initiative (QBOi) to improve the simulation of tropical stratospheric variability in General Circulation Models and Earth System Models (GCMs and ESMs). While QBOi is focused on modelling studies, it is also closely aligned with other SPARC activities including the SPARC Reanalysis Intercomparison Project (S-RIP; Fujiwara et al., 2017) providing supporting analysis of observations and reanalyses, and with the SPARC gravity waves activity (Alexander and Sato, 2015) that is studying an important driver of the QBO.

Unlike the Coupled Model Intercomparison Project Phase 6 (CMIP6; Eyring et al., 2016), and to a lesser extent CCMI, the design of experiments for QBOi is not governed by the huge and rather diverse requirements from policy makers and scientists that has presented such a massive cultural and organizational challenge to the modelling community (Eyring et al., 2016). Instead, QBOi has adopted a less onerous approach for experimental setup using stand-alone experiments (Section 3) specifically focused on improving the representation of the QBO in GCMs and addressing scientific questions related to advancing understanding of the QBO per se (Section 2). This is an essential prerequisite to improving the representation in models of important QBO influences (Baldwin et al., 2001) such as the modulation of the transport of aerosols and chemical constituents into and within the stratosphere (e.g., Strahan et al., 2015) or the dynamical teleconnections to the extra-tropics (Anstey and Shepherd, 2014) and their subsequent surface climate and weather impacts. These aspects are expected to be included more prominently in the next phase of QBOi. The purpose of this paper is to describe the experiments to be used in phase 1 of QBOi and provide supporting documentation for other publications analysing and interpreting the output from the experiments. To help promote widest possible participation in the experiments, and thereby maximize the size of the multi-model ensembles,





**Figure 1.** Ten-year (1990-1999) time series of monthly and zonal-mean zonal wind at the equator from 100 hPa to 10 hPa for 47 models that uploaded data to the Coupled Model Intercomparison Project Phase 5 (CMIP5) data repository. Only five models (CMCC-CMS, HadGEM2-CC, MIROC-ESM, MIROC-ESM-CHEM, and MPI-ESM-MR) spontaneously produce the iconic QBO behaviour of alternating descending layers of eastward and westward winds such as indicated in the upper left hand panel for ERA-Interim reanalysis (Dee et al., 2011). Equatorial stratospheric winds in the CMIP5 version of CESM1(WACCM) are strongly relaxed ("nudged") toward observations, which is why it shows a close resemblance to ERA-Interim in this figure. (Note that the version of WACCM participating in QBOi, described in Section 5, is a different version of this model.)

the design of the experiments has involved input from the community throughout (Anstey et al., 2015; Hamilton et al., 2015). The scientific rationale for the experiments also evolved through community discussion (Anstey et al., 2015) and is presented in the next section.

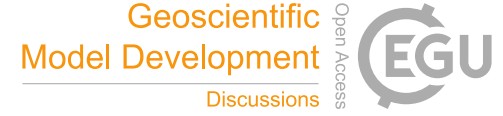

An important part of the multi-model analysis and interpretation of the experiments is the availability of a consistent set of relevant diagnostics from each model. For this QBOi follows best practices and, where possible, variable and file naming conventions of CMIP5 and CCMI (see Supplement). In particular the recommended output diagnostics are adapted from those requested by the Dynamics and Variability Model Intercomparison Project (DynVarMIP; Gerber and Manzini, 2016a).

These will allow for the zonal-mean zonal momentum budgets to be examined in detail in the Transformed Eulerian Mean (TEM) framework (e.g., Andrews et al., 1987, p. 127–130) including contributions from parameterized (sub grid-scale) gravity waves. Other requested diagnostics are aimed at characterizing the sources, propagation and filtering (i.e., breaking) of both resolved and unresolved waves in the participating models, particularly in the equatorial region. Precise specification of the requested diagnostics can be found in Section 4. To facilitate the comparison of these diagnostics among participating models,

salient model features that are important for capturing QBO-like behavior are described and tabulated in Section 5, with some emphasis in particular on the non-orographic gravity wave drag (GWD) parameterizations used by almost all of the QBOi models. Closing remarks including future plans follow in Section 6.

## 2   Scientific rationale

A crucial test of our understanding and ability to model the QBO occurred around the beginning of 2016 when the QBO cycle

was unexpectedly disrupted for the first time since its discovery in the late 1950s (Dunkerton et al., 2016; Newman et al., 2016; Osprey et al., 2016; Coy et al., 2017). The well established QBO paradigm, originating from the 1960s, of alternate eastward and westward momentum deposition from vertically propagating equatorial waves (Baldwin et al., 2001) could not account for this disruption (Osprey et al., 2016). Despite the fact that the QBO is normally highly predictable (Pohlmann et al., 2013; Scaife et al., 2014) the disruption was completely missed by seasonal forecasts, and this failure illustrates the difficulty models

have in capturing the complex phenomenology of the QBO and its full range of variability. Similar disruptions have only very rarely been seen in multi-decadal simulations and from just a few models with QBO-like oscillations (e.g., Osprey et al., 2016). It is possible that the models may be over-tuned to ensure that they capture the mean behaviour of selected metrics (e.g., mean period and amplitude) of the present-day QBO. Furthermore, the disruption itself raises the possibility that the real QBO is less robust than previously thought, although it has since returned to its usual cycling as predicted.

With the advent of non-orographic GWD parameterizations and/or the use of increased vertical resolution in the stratosphere, a growing number of global models have been able to reproduce QBO-like variability in the equatorial stratosphere (e.g., Takahashi, 1996; Scaife et al., 2000; Hamilton et al., 2001; Giorgetta et al., 2002; Shibata and Deushi, 2005; Anstey et al., 2010; Kawatani et al., 2010; Orr et al., 2010; Lott and Guez, 2013; Richter et al., 2014; Rind et al., 2014; McCormack et al., 2015). However, common deficiencies exist in all current simulations, notably with QBO winds often being unrealistically

weak in the lowermost stratosphere and having unrealistically small cycle-to-cycle variability (e.g., Schenzinger et al., 2017). The simulated QBOs can also be quite "fragile"– which is to say, sensitive to many different aspects of model formulation depending on the model. For example the QBO in the Canadian Middle Atmosphere Model (AGCM3-CMAM) is sensitive to the balance of resolved and parameterized wave forcing (Anstey et al., 2016) while in different versions of the Met Office



Unified Model (MetUM) the QBO is sensitive to the specification of stratospheric ozone (Butchart et al., 2003; Bushell et al., 2010) and/or the parameterized gravity waves (Bushell et al., 2010; Kim et al., 2013). Sensitivity to vertical resolution has been reported by numerous studies, for example by Giorgetta et al. (2006) for the Middle Atmosphere version of the ECHAM5 (MAECHAM5) model and by Geller et al. (2016) for the NASA Goddard Institute for Space Studies (GISS) climate model. In

addition Yao and Jablonowski (2015) identified a sensitivity to the choice of dynamical core. Other key questions concerning simulation of the QBO lie with its possible synchronisation with other modes of variability such as the annual cycle (e.g., Rajendran et al., 2016) and El Niño-Southern Oscillation (e.g., Christiansen et al., 2016), with the QBO's predictability (e.g., Pohlmann et al., 2013; Scaife et al., 2014) and finally with the robustness of the QBO response to climate change (e.g. Kawatani and Hamilton, 2013; Schirber et al., 2015).

Phase 1 of QBOi focuses on reducing these uncertainties in simulated QBOs by conducting coordinated experiments that will allow for more rigorous intercomparison of models than is otherwise possible from individual studies. The aim is to address the ability of GCMs to capture the QBO in the present climate, to predict its behaviour under climate-change forcings, and to predict its evolution when initialized with observations (i.e., hindcasts).

## 3  Experiments

Anstey et al. (2015) and Hamilton et al. (2015) briefly describe a set of five QBO experiments which are designed to be simple and accessible to a wide range of modelling groups. The motivation and specific goals for each of these experiments is presented below with the technical specifications given in Appendix A. The aim is for modelling groups to perform all five experiments and even if this is not possible, it is important that the same model version is used for the subset of experiments that are conducted, i.e., there should be no tuning of free parameters between experiments. Use of the same model version for

the different experiments is crucial for learning the most from this study. The model version used should be that which the group considered gave the "best" representation of the QBO under present day conditions (e.g., in Experiment 1 or similar preparatory simulations). Of course there are situations when two different versions of a model might be used to perform the experiment set, such as when high and low resolution versions or alternative non-orographic GWD parameterizations are available. In these situations the results would then be treated for the purpose of the QBOi multi-model analysis as if they were

obtained from two separate models (although interpretation of results will need to be aware of, and test for sensitivity to, the possible dominance of the results by one particular family of models). All experiments are for AGCMs apart for an option to perform Experiment 5 with a coupled ocean, which is denoted as Experiment 5A (see below).

### 3.1  Experiment list and goals

#### 3.1.1  Present-day climate

The first two experiments are designed with the goal of identifying and distinguishing the properties of and mechanisms underlying the variety of model simulations of the QBO in present-day conditions:





- **Experiment 1 ("AMIP"):** Specified observed interannually varying sea surface temperatures (SSTs), sea ice and external forcings for $1^{st}$ January 1979 to $28^{th}$ February 2009 (1-3 member ensemble).

- **Experiment 2 (present-day timeslice):** Identical to Experiment 1 except employing repeated annual cycle SSTs, sea ice, and external forcings (100 years or ensemble of 3×30 years).

5    The main differences between these two experiments are expected to arise from the differences between their specified SSTs. Figure 2 compares the variability in the tropics (5°N - 5°S) of the prescribed SSTs for Experiments 1 and 2. Averaged over all longitudes the differences are relatively small (Figure 2, top panel), although regionally there are large differences, for instance due to the effects of the El Niño-Southern Oscillation (Figure 2, bottom panel).

These experiments will allow an evaluation of the accuracy of modelled QBOs under present-day climate conditions, em-
10  ploying the diagnostics and metrics discussed in Section 4. The impact of interannually varying forcing (e.g., Figure 2) on the model QBO will be assessed through a comparison of the two experiments. Experiment 2 also provides the control for the climate projection Experiments 3 and 4.

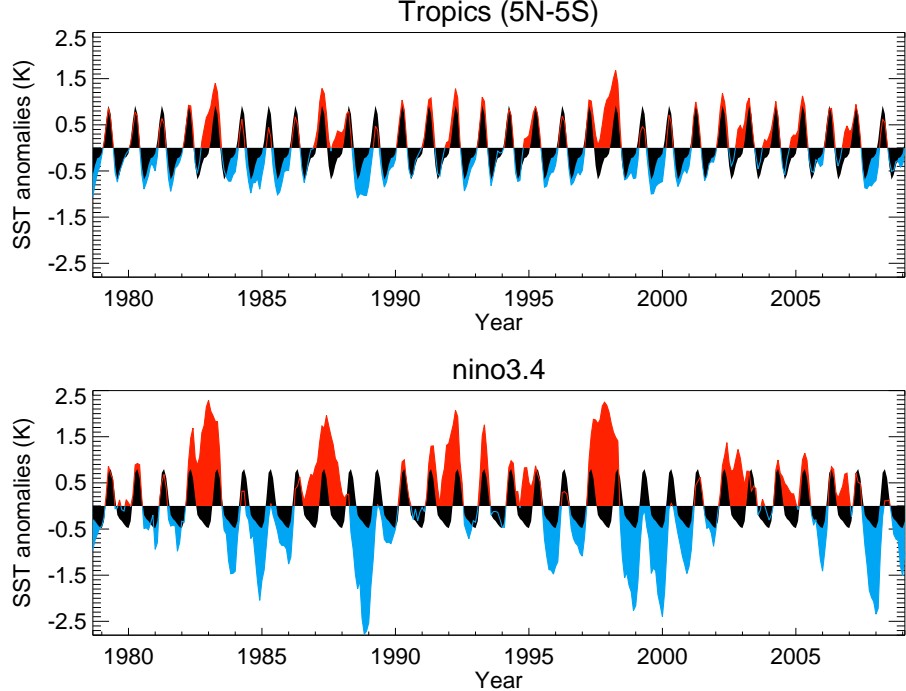

**Figure 2.** Comparison of monthly mean sea surface temperature (SST) anomalies (red and blue) from the 30 year mean (1979-2008) of the CMIP5 AMIP SSTs used in Experiment 1 with the mean annual cycle (black) for the same period. Top panel: average for all longitudes between 5°N and 5°S. Bottom panel: average for the Niño 3.4 region (120°-170°W, 5°N-5°S).



### 3.1.2 Climate projections

Two further experiments are designed to subject the modelled QBOs (i.e., the QBO simulated by the present-day experiments) to an external forcing similar to that typically applied for climate projections:

- **Experiment 3 ($2\times CO_2$ timeslice):** Identical to Experiment 2, but with a change in $CO_2$ concentration and specified SSTs appropriate for a $2\times CO_2$ world (100 years or ensemble of $3\times30$ years).

- **Experiment 4: ($4\times CO_2$ timeslice):** Identical to Experiment 2, but with a change in $CO_2$ concentration and specified SSTs appropriate for a $4\times CO_2$ world (100 years or ensemble of $3\times30$ years).

These experiments will allow the response (i.e., $2\times CO_2$ - $1\times CO_2$ and $4\times CO_2$ - $1\times CO_2$) of the QBO, its forcing mechanisms, and its impact/influence to be evaluated using the same diagnostics and metrics used in the analysis of Experiments 1 and 2. Key questions that will be addressed are:

- What is the spread/uncertainty of the forced model response?

- Do different models cluster in any particular way?

- Can a connection/correlation be made between QBOs with similar metrics/diagnostics in present-day climate and their response to $CO_2$ forcing?

The motivation is to investigate what aspects of modelled QBOs determine the spread, or uncertainty, of the QBO response to $CO_2$ forcing. These aspects are considered high priority by QBOi in order to reduce uncertainty in future projections. These experiments also will provide context for the uncertainty in climate change projections of QBO behaviour among the state-of-the-art GCMs being used in CMIP6.

Furthermore, the possibility was noted in Section 2 that models may be over-tuned to ensure that they capture the behaviour of the present-day QBO. If so, then a large multi-model spread in the forced response may indicate that such tuning constitutes, in effect, an "overfitting" of models to present-day conditions.

### 3.1.3 QBO hindcasts

The goal of the final experiment is to evaluate and compare the predictive skill of modelled QBOs in a retrospective hindcast context, quantify this predictive capability in multiple models, and study the model processes driving the evolution of the QBO:

- **Experiment 5 (hindcasts):** A set of initialized QBO hindcasts of 9-12 months using the observed SSTs and forcings specified as in Experiment 1. Specified start dates are $1^{st}$ May and $1^{st}$ November for the years 1993-2007 (i.e., 15 years, 30 start dates) with initial atmospheric conditions obtained from reanalyses (at least 3-member ensemble).

Because of the prescribed SSTs these are not true prediction experiments; nonetheless they provide an important test of how well models can predict the evolution of the QBO from specified initial conditions that reasonably sample the full range of QBO phases, despite some clustering of the $1^{st}$ May initial profiles (Figure 3).





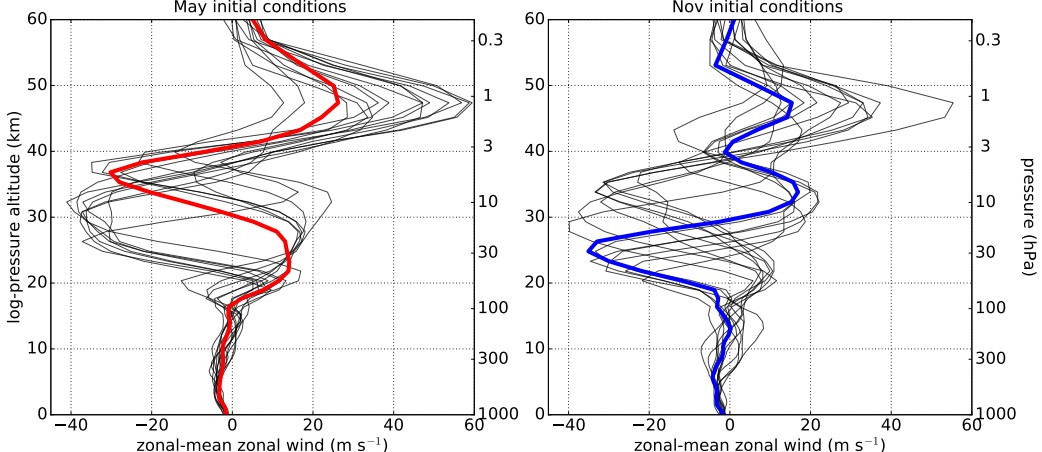

**Figure 3.** Zonal-mean and daily-mean zonal wind (m s$^{-1}$) profiles at the equator for the $1^{st}$ May and $1^{st}$ November for the 15 years 1993-2007, from ERA-Interim reanalyses (Dee et al., 2011). The two profiles shown in coloured lines (May 1993 and November 2005, taken as representative of eastward and westward QBO phases in the lower stratosphere, respectively) are those used in offline comparison of the gravity-wave drag parameterizations presented in Section 5.1.

Key questions that will be addressed are:

– How does prediction skill vary among models, and to what extent, and for how long are models able to predict the QBO evolution correctly at different vertical levels and different phases of the QBO?

– How does the forecast skill relate to the behaviour of the QBO in Experiment 1? Are realistic QBO simulations in a multi-decadal simulation well correlated with skillful long-term deterministic predictions?

– Do the models that cluster and/or do well in the prediction experiments cluster in the $CO_2$ forcing experiments?

One aim is to investigate which aspects of modelled QBOs determine the quality of QBO prediction and therefore where development needs to be focused for model improvement. The hindcast framework can also be helpful for directly assessing model changes, possibly driving improvements in free-running models. Further motivation for these experiments is to investigate the possibility of using the hindcast results to narrow the range of plausible models for climate change experiments.

It is recognised that some groups may already have completed for the period 1993-2007 operational seasonal hindcasts using a coupled ocean-atmosphere model, and therefore for the QBOi multi-model analysis an acceptable alternative (or addition) to Experiment 5 is:

– **Experiment 5A (hindcasts):** A set of initialized QBO hindcasts of 9-12 months identical to Experiment 5 apart from replacing the specified SSTs with a coupled ocean model appropriately initialized (at least 3 member ensemble).

Full comparison with the other models providing Experiment 5 output will nonetheless depend on most of the diagnostics discussed in Section 5 being available from those groups providing Experiment 5A output.

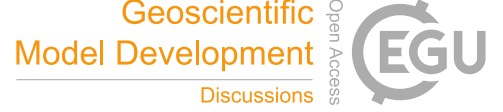



### 3.1.4 Process studies

A secondary purpose of Experiment 5 is to investigate and evaluate differences in wave dissipation and momentum deposition, so as to understand the processes driving the QBO in each model and separate the contributions from resolved and unresolved waves (e.g., Scaife et al., 2000; Shibata and Deushi, 2005). Due to the initialization of the hindcasts, each model will have essentially the same initial basic state, and its evolution immediately after the start of the forecast will allow the properties of wave dissipation and momentum deposition to be compared and contrasted between different models given a near-identical basic state. Specifying the same observed SST in all models (rather than allowing each model to predict its own SST evolution) facilitates the comparison as it eliminates any differences resulting from the evolving ocean. Short periods of additional high frequency diagnostics are requested to maximize the benefits of the multi-model comparison.

## 4  Diagnostics

The diagnostics requested by QBOi draw on those requested by other major multi-model intercomparison projects, in particular DynVarMIP (Gerber and Manzini, 2016a), though they have been specifically tailored through community discussion for the analysis of the QBO in Experiments 1-5. The requested diagnostics are described in this section; additional technical information on how they should be formatted and uploaded to the shared QBOi repository is available in the Supplement.

### 4.1  Spatial and temporal resolution

For ease of comparison among models most output variables are requested on a standard set of 30 pressure levels: 1000, 925, 850, 700, 600, 500, 400, 300, 250, 200, 175, 150, 120, 100, 85, 70, 60, 50, 40, 30, 20, 15, 10, 7, 5, 3, 2, 1.5, 1.0 and 0.4 hPa. These are adapted from the extended levels set requested by DynVarMIP for CMIP6 (e.g., Gerber and Manzini, 2016a) to obtain a vertical resolution in the upper tropical troposphere and lower stratosphere (i.e., between 200 hPa and 40 hPa) of 1.0 to 1.5 km. There are two exceptions however:

- Data to be used for calculating equatorial wave spectra (6-hourly instantaneous fields) should be provided at vertical resolution equivalent to the model resolution to ensure accurate calculation of QBO wave forcing (e.g., Kim and Chun, 2015a); see below for further details.

- To reduce data volume, daily-mean 3-dimensional (3D) variables are requested for only the 8 pressure levels used by CMIP5: 1000, 850, 700, 500, 250, 100, 50 and 10 hPa. These data will be used mainly to examine the QBO influence on other regions of the atmosphere [e.g., on the North Atlantic Oscillation (NAO)] and higher vertical resolution is not considered necessary.

Horizontal resolution should be the same as the model but if data volume is an issue then a reduced grid is acceptable, provided the reduction method is documented.

To examine the daily-mean and monthly-mean QBO zonal-mean momentum budget, terms making up the TEM zonal momentum equation (e.g., Andrews et al., 1987, p. 127–130) are requested following the recipe given by Gerber and Manzini



(2016a, Appendix A3), but also see their *corrigendum* (Gerber and Manzini, 2016b). In particular note the importance of calculating the individual terms from 6 hourly or higher frequency data (e.g., every time step) and the need for sufficient vertical resolution (e.g., the standard pressure levels listed above) for accurate estimates of the vertical derivatives. Furthermore to examine the wavenumber-frequency spectra of the equatorial waves (e.g., Horinouchi et al., 2003; Lott et al., 2014) instantaneous

values of 3D winds and temperature are requested every 6 hours on model levels or on pressure levels at roughly equivalent vertical resolution to the model levels but, to reduce data volumes, only for levels between 100 hPa and 0.4 hPa and for latitudes between 15°N and 15°S. For ease of analysis, pressure levels at model-level resolution are preferred over actual model levels.

**Table 1.** Climate and variability. Monthly and daily means, with 2D indicating a longitude-latitude-time (XYT) field and 3D indicating a longitude-latitude-pressure-time (XYPT) field. XY is typically the model's horizontal output grid and P is the standard 30-level set of diagnostic pressure levels described in Section 4.1: 1000, 925, 850, 700, 600, 500, 400, 300, 250, 200, 175, 150, 120, 100, 85, 70, 60, 50, 40, 30, 20, 15, 10, 7, 5, 3, 2, 1.5, 1.0 and 0.4 hPa.

| Name | Long name [units] | Dimension |
|------|-------------------|-----------|
| psl | sea level pressure [Pa] | 2D |
| prc | convective precipitation flux [$\mathrm{kg\,s^{-1}m^{-2}}$] | 2D |
| pr | total precipitation flux [$\mathrm{kg\,s^{-1}m^{-2}}$] | 2D |
| tas | near-surface air temperature [K] | 2D |
| uas | eastward near-surface wind [$\mathrm{m\,s^{-1}}$] | 2D |
| vas | northward near-surface wind [$\mathrm{m\,s^{-1}}$] | 2D |
| ta | air temperature [K] | 3D* |
| ua | eastward wind [$\mathrm{m\,s^{-1}}$] | 3D* |
| zg | geopotential height [m] | 3D* |

*For daily 3D variables P is reduced to 8 pressure levels: 1000, 850, 700, 500, 250, 100, 50, 10 hPa.

## 4.2 Output period

Monthly-mean output is requested for the full duration of all experiments and all ensemble members. Likewise for Experiment

5 daily-mean output is requested for the full duration of each ensemble member. On the other hand for Experiments 1-4 daily-mean output is only requested for the first 30 years and/or the first ensemble member.

High-frequency (6-hourly) diagnostics for calculating equatorial wave spectra are requested for the following periods and ensemble members for each experiment:

- Experiment 1: 1997-2002 [note this period encompasses positive, negative and neutral El Niño-Southern Oscillation

(ENSO) phases] of first ensemble member

- Experiments 2-4: years 1-4 of first ensemble member

- Experiment 5: first 3 months of all ensemble members





**Table 2.** Dynamics. (a) Monthly-mean and daily-mean fields and contributions to zonal-mean zonal momentum equation (YPT). (b) Monthly-mean tendencies and fluxes from parameterized gravity waves (XYPT). (c) Daily-mean sources for orographic and non-orographic gravity waves (XYT). P is the standard 30-level set of diagnostic pressure levels described in Section 4.1 (also Table 1 caption).

| (a) Monthly-mean & daily-mean, zonal-mean fields - YPT | | |
| --- | --- | --- |
| Name | Long name [units] | Dimension |
| ua | eastward wind [$\mathrm{m\,s^{-1}}$] | 2D |
| ta | air temperature [K] | 2D |
| zg | geopotential height [m] | 2D |
| vstar | residual northward wind [$\mathrm{m\,s^{-1}}$] | 2D |
| wstar | residual upward wind [$\mathrm{m\,s^{-1}}$] | 2D |
| fy | northward EP-flux [$\mathrm{N\,m^{-1}}$] | 2D |
| fz | upward EP-flux [$\mathrm{N\,m^{-1}}$] | 2D |
| utenddivf | u-tendency by EP-flux divergence [$\mathrm{m\,s^{-2}}$] | 2D |
| utend | u-tendency [$\mathrm{m\,s^{-2}}$] | 2D |
| utendogw | u-tendency by orographic gravity waves [$\mathrm{m\,s^{-2}}$] | 2D |
| utendnogw | u-tendency by non-orographic gravity waves [$\mathrm{m\,s^{-2}}$] | 2D |
| psistar | residual stream function [$\mathrm{kg\,s^{-1}}$] | 2D |
| (b) Monthly-mean gravity wave tendencies and fluxes - XYPT | | |
| utendogw | u-tendency by orographic gravity waves [$\mathrm{m\,s^{-2}}$] | 3D |
| utendnogw | u-tendency by non-orographic gravity waves [$\mathrm{m\,s^{-2}}$] | 3D |
| vtendogw | v-tendency by orographic gravity waves [$\mathrm{m\,s^{-2}}$] | 3D |
| vtendnogw | v-tendency by non-orographic gravity waves [$\mathrm{m\,s^{-2}}$] | 3D |
| taunoge | eastward wind stress of non-orographic gravity waves [Pa] | 3D |
| taunogs | southward wind stress of non-orographic gravity waves [Pa] | 3D |
| taunogw | westward wind stress of non-orographic gravity waves [Pa] | 3D |
| taunogn | northward wind stress of non-orographic gravity waves [Pa] | 3D |
| (c) Daily-mean gravity wave sources - XYT | | |
| tauogu | surface eastward wind stress by orographic gravity waves [Pa] | 2D |
| tauogv | surface northward wind stress by orographic gravity waves [Pa] | 2D |
| taunoge[†] | launch eastward wind stress of non-orographic gravity waves [Pa] | 3D |
| taunogs[†] | launch southward wind stress of non-orographic gravity waves [Pa] | 3D |
| taunogw[†] | launch westward wind stress of non-orographic gravity waves [Pa] | 3D |
| taunogn[†] | launch northward wind stress of non-orographic gravity waves [Pa] | 3D |

[†] only if non-isotropic and/or non-stationary at launch-level (e.g., coupled to convection or fronts).





## 4.3 Requested output variables

Similarly to DynVarMIP (Gerber and Manzini, 2016a), the requested variables are separated into three categories: standard variables (Table 1) for diagnosing the climate and variability in the models, dynamical variables (Table 2) for analysing momentum transport and budgets, and thermodynamic quantities (Table 3). In addition a fourth category of variables (Table 4) will enable the equatorial wave spectra (e.g., Horinouchi et al., 2003; Lott et al., 2014) to be compared among the models.

**Table 3.** Thermodynamics. Monthly-mean and daily-mean zonal-mean fields (YPT). P is the standard 30-level set of diagnostic pressure levels described in Section 4.1 (also Table 1 caption).

| (a) Monthly-mean & daily-mean, zonal-mean fields - YPT | | |
|---|---|---|
| Name | Long name [units] | Dimension |
| hus | specific humidity [$kg\,kg^{-1}$] | 2D |
| zmtnt | diabatic heating rate [$K\,s^{-1}$] | 2D |
| tntlw | longwave heating rate [$K\,s^{-1}$] | 2D |
| tntsw | shortwave heating rate [$K\,s^{-1}$] | 2D |
| o3$^{\$}$ | mole fraction of ozone in air [$mole\,mole^{-1}$] | 2D |

$^{\$}$only if model has prognostic ozone.

**Table 4.** Equatorial wave spectra. Six hourly instantaneous 3D (XYPT) equatorial fields (15°N to 15°S) output for selected sub-periods of each experiment (see Section 4.2). Here P is *not* the standard set of pressure levels used in Tables 1–3. Rather, as described in Section 4.1, it indicates a set of pressure levels with equivalent vertical resolution to the model levels, covering the altitude range 100 to 0.4 hPa. Alternatively the data can be provided on actual model levels, although in this case the data required for conversion between model and pressure levels must also be provided.

| Six-hourly equatorial fields - XYPT | | |
|---|---|---|
| Name | Long name [units] | Dimension |
| ta | air temperature [K] | 3D |
| ua | eastward wind [$m\,s^{-1}$] | 3D |
| va | northward wind [$m\,s^{-1}$] | 3D |
| wa | vertical wind [$m\,s^{-1}$] | 3D |

## 5 Participating models

All the experiments for phase 1 of QBOi have been designed for atmosphere-only GCMs. From the experiment descriptions in Section 3 it is also clear that for an AGCM to participate in these experiments it must be configured with a number of essential characteristics (e.g., land-ocean contrast, annual cycle, and a radiation scheme that can accommodate changes in $CO_2$ amounts).





**Table 5.** Participating models and contact information.

| Model names | Expts. | Institutes | Investigators | Email address | References |
|---|---|---|---|---|---|
| 60LCAM5 | 1-4 | NCAR | J. Chen | cchen@ucar.edu | Richter et al. (2014) |
| | | | J. Richter | jrichter@ucar.edu | |
| AGCM3-CMAM | 1-3, 5 | CCCMa | J. Anstey | james.anstey@canada.ca | Scinocca et al. (2008) |
| | | | J. Scinocca | john.scinocca@canada.ca | Anstey et al. (2016) |
| | | U. Toronto | C. McLandress | charles@atmosp.physics.utoronto.ca | |
| CESM1- | 1-4 | NCAR | R. Garcia | rgarcia@ucar.edu | |
| (WACCM-L110) | | | J. Richter | jrichter@ucar.edu | Garcia and Richter (2017) |
| EC-EARTH3.1 | 5 | BSC | J. Garcia-Serrano | javier.garcia@bsc.es | Christiansen et al. (2016) |
| ECHAM5sh | 1-4 | ISAC-CNR | F. Serva | federico.serva@artov.isac.cnr.it | Serva et al. (2017) |
| | | | C. Cagnazzo | c.cagnazzo@isac.cnr.it | Manzini et al. (2012) |
| EMAC | 1-4 | KIT | P. Braesicke | peter.braesicke@kit.edu | Jöckel et al. (2005) |
| | | | T. Kerzenmacher | tobias.kerzenmacher@kit.edu | Jöckel et al. (2010) |
| | | | S. Versick | stefan.versick@kit.edu | |
| HadGEM2-A | 1 | Ewha W. U. | Y.-H. Kim | young-ha.kim@ewha.ac.kr | Martin et al. (2011) |
| | | Yonsei U. | H.-Y. Chun | chunhy@yonsei.ac.kr | |
| HadGEM2-AC | 1 | Ewha W. U. | Y.-H. Kim | young-ha.kim@ewha.ac.kr | Martin et al. (2011) |
| | | Yonsei U. | H.-Y. Chun | chunhy@yonsei.ac.kr | Kim and Chun (2015b) |
| IFS43r1 | 1-5 | ECMWF | T. Stockdale | tim.stockdale@ecmwf.int | ECMWF (2016); Orr et al. (2010) |
| LMDz6 | 1-4 | ISPL-LMD | F. Lott | flott@lmd.ens.fr | Lott et al. (2005, 2012) |
| MIROC-AGCM-LL | 1-5 | MIROC | Y. Kawatani | yoskawatani@jamstec.go.jp | Kawatani et al. (2011) |
| MIROC-ESM | 1-5 | MIROC | S. Watanabe | wnabe@jamstec.go.jp | Watanabe et al. (2011) |
| MPI-ESM-MR | 5A | MPI | H. Pohlmann | holger.pohlmann@mpimet.mpg.de | Pohlmann et al. (2013) |
| | | U. Hamburg | M. Dobrynin | mikhail.dobrynin@uni-hamburg.de | Dobrynin et al. (2016) |
| MRI-ESM2 | 1-5 | MRI-JMA | K. Yoshida | kyoshida@mri-jma.go.jp | Adachi et al. (2013) |
| | | | H. Naoe | hnaoe@mri-jma.go.jp | Yukimoto et al. (2012) |
| | | | S. Yukimoto | yukimoto@mri-jma.go.jp | |
| UMGA7 | 1-4 | Met Office | A. Bushell | andrew.bushell@metoffice.gov.uk | Walters et al. (2016) |
| | | MOHC | N. Butchart | neal.butchart@metoffice.gov.uk | |
| | | U. Oxford | S. Osprey | scott.osprey@physics.ox.ac.uk | |
| UMGA7gws | 1-4 | Met Office | A. Bushell | andrew.bushell@metoffice.gov.uk | Bushell et al. (2015) |
| | | MOHC | N. Butchart | neal.butchart@metoffice.gov.uk | Walters et al. (2016) |
| | | U. Oxford | S. Osprey | scott.osprey@physics.ox.ac.uk | |
| UMGC2 | 5A | MOHC | A. Scaife | adam.scaife@metoffice.gov.uk | Dunstone et al. (2016) |



Apart from this QBOi does not impose any restrictions on the representation in participating models of any physical process or, indeed, chemical process for those models with interactive ozone. Of course, participating models are expected to properly resolve the stratosphere with an average vertical resolution of the order of 2 km or less between 100 hPa and 1 hPa and an upper boundary somewhere above that (cf., high and low top results in Osprey et al., 2013). However, it is not strictly necessary for a

5 model to display QBO-like variability in the equatorial stratosphere as additional insight can be gained by comparing models with and without this property. Models with QBO-like variability but without a properly resolved stratosphere (e.g., with upper boundary below 1 hPa) are also considered since, again, this potentially provides guidance on the level of stratospheric detail that is required in order to reproduce a QBO. There are 17 models or model-versions participating in phase-1 of QBOi (i.e.,

**Table 6.** Model domain and resolution

| Model name | Horizontal resolution | No. of levels | Upper boundary | Timestep |
|---|---|---|---|---|
| 60LCAM5 | 100 km | 60 | 2.5 hPa (41 km) | 30 min |
| AGCM3-CMAM | T47 | 113 | 0.00074 hPa (98 km) | 7.5 min |
| CESM1(WACCM5-110L) | $1.25° \times 0.94°$ | 110 | $6.1 \times 10^{-6}$ hPa (132 km) | 30 min |
| EC-EARTH3.1 | T255 | 91 | 0.01 hPa (80 km) | 45 min |
| ECHAM5sh | T63 | 95 | 0.01 hPa (80 km) | 7.5 min |
| EMAC | T42 | 90 | 0.01 hPa (80 km) | 12 min |
| HadGEM2-A | $1.875° \times 1.25°$ | 60 | 0.006 hPa (84 km) | 20 min |
| HadGEM2-AC | $1.875° \times 1.25°$ | 60 | 0.006 hPa (84 km) | 20 min |
| IFS43r1 | T255 | 137 | 0.01 hPa (80 km) | 30 min |
| LMDz6 | $2.5° \times 1.25°$ | 79 | 0.015 hPa (77 km) | 3 min |
| MIROC-AGCM-LL | T106 | 72 | 1.2 hPa (47 km) | 5 min |
| MIROC-ESM | T42 | 80 | 0.0036 hPa (87 km) | 30 min |
| MPI-ESM-MR | T63 | 95 | 0.01 hPa (80 km) | 7.5 min |
| MRI-ESM2 | T159 | 80 | 0.01 hPa (80 km) | 30 min |
| UMGA7 | $1.875° \times 1.25°$ | 85 | 0.0053 hPa (85 km) | 20 min |
| UMGA7gws | $1.875° \times 1.25°$ | 85 | 0.0053 hPa (85 km) | 20 min |
| UMGC2 | $0.833° \times 0.556°$ | 85 | 0.0053 hPa (85 km) | 15 min |

For spectral models the horizontal resolution is given in terms of triangular truncation of spectral coefficients, from which a grid spacing can be estimated as described in the Figure 5 caption. For example, T63 $\sim 2.8° \times 2.8°$, T159 $\sim 1.125° \times 1.125°$, and T255 $\sim 0.7° \times 0.7°$, corresponding roughly to grid lengths 310 km, 130 km and 80 km, respectively. Upper boundary altitude is given in terms of pressure and log-pressure altitude as described in the Figure 4 caption.

data from 17 models has been uploaded or is planned for upload to the shared QBOi repository; see Supplement for details

10 of this repository). These models are listed in Table 5 along with the institutes and investigators using the models and their contact information. The model names given refer to the names used in the repository while the information given in Tables





**Table 7.** Non-orographic gravity waves, convection and ozone chemistry

| Model name | Non-orographic gravity waves | Non-orographic GW source | Convection | Ozone chemistry |
|---|---|---|---|---|
| 60LCAM5 | Lindzen (1981)[‡] | Richter et al. (2010) | Zhang and McFarlane (1995) | no |
| AGCM3-CMAM | Scinocca (2003)[†] [WM] | fixed | Zhang and McFarlane (1995) | no |
| CESM1(WACCM5-110L) | Lindzen (1981)[‡] | Richter et al. (2010) | Zhang and McFarlane (1995) | no |
| EC-EARTH3.1 | Scinocca (2003)[†] [WM] | fixed | Davini et al. (2017) | no |
| ECHAM5sh | Hines (1997a, b)[†] [H] | fixed | Tiedtke (1989), Nordeng (1994) | no |
| EMAC | Hines (1997a, b)[†] [H] | fixed | Tiedtke (1989) | no |
| HadGEM2-A | Warner and McIntyre (1999)[†] [WM] | fixed | Gregory et al. (1990) | no |
| HadGEM2-AC | Warner and McIntyre (1999)[†] [WM] Warner and McIntyre (1999)[†] [WM] | fixed Choi and Chun (2011) | Gregory et al. (1990) | no |
| IFS43r1 | Scinocca (2003)[†] [WM] | fixed | Bechtold et al. (2008) | yes |
| LMDz6 | Lott et al. (2012)[‡] [L] | de la Cámara and Lott (2015), Lott and Guez (2013) | Emanuel (1991), Hourdin et al. (2013) | no no |
| MIROC-AGCM-LL | none | — | Emori et al. (2001) | no |
| MIROC-ESM | Hines (1997a, b)[†] [H] | fixed | Emori et al. (2001) | no |
| MPI-ESM-MR | Hines (1997a, b)[†] [H] | fixed | Tiedtke (1989), Nordeng (1994) | no |
| MRI-ESM2 | Hines (1997a, b)[†] [H] | fixed | Yoshimura et al. (2015) | yes |
| UMGA7 | Warner and McIntyre (1999)[†] [WM] | fixed | Gregory et al. (1990) | no |
| UMGA7gws | Warner and McIntyre (1999)[†] [WM] | Bushell et al. (2015) | Gregory et al. (1990) | no |
| UMGC2 | Warner and McIntyre (1999)[†] [WM] | fixed | Gregory et al. (1990) | no |

Schemes marked [†] are non-orographic GWD parameterizations based on a wave-spectrum approach, while in schemes marked [‡] the wave spectrum is treated as a collection of monochromatic waves. For the models using the Warner and McIntyre (1999) scheme (HadGEM2-A, HadGEM2-AC, UMGA7, UMGA7gws, and UMGC2), the use of the scheme to generate a QBO is described in Scaife et al. (2002). For IFS43r1, the use of the Scinocca (2003) scheme to generate a QBO is described in Orr et al. (2010). For MPI-ESM-MR, the use of the Hines (1997a, b) scheme is described in Schmidt et al. (2013). The abbreviation in square brackets for each scheme (2nd column; "[WM]", "[H]" or "[L]") denotes the type of dissipation used in the scheme as labelled in Figure 7. "Fixed" in column 3 refers to sources of parameterized gravity-waves that are not linked to any other model physical variable (see Footnote 1, Section 5). Note however that "fixed" includes sources that vary in time and/or space in a prescribed way, as well as stochastically (e.g. as is done in the ECHAM5sh model).

6 and 7 refers specifically to the configuration and parameter settings used by each model when producing the uploaded data. More comprehensive descriptions of the individual models can be found in the references given in the last column of Table 5.

It should be noted that common model development history can lead to a lack of full independence among models. For example, 60LCAM5 and CESM1(WACCM5-110L) have developed from the NCAR Community Atmosphere Model (CAM); HadGEM2-A, HadGEM2-AC, UMGA7, UMGA7gws and UMGC2 have developed out the Met Office Unified Model (UM); EC-EARTH3.1 and IFS have their origins in the ECMWF Integrated Forecasting System (IFS); MIROC-AGCM and MIROC-ESM belong to the family of MIROC models; and ECHAM5sh, EMAC and MPI-ESM-MR all originate from the MPI ECHAM




line of model development. Tables 6 and 7 indicate that some model components are shared by different models. The extent to which shared development history affects model independence can be difficult to assess and varies among models (e.g., Knutti et al., 2013). Apart from describing those aspects of model formulation that are expected to be relevant to the QBO (Tables 6 and 7), detailed consideration of model independence is outside the scope of this paper. However, note that out of the 17

QBOi models, there are two pairs of models that are identical in all respects but one: HadGEM2-A and UMGA7 used fixed sources for their non-orographic gravity wave parameterizations, while their counterparts HadGEM2-AC and UMGA7gws, respectively, use parameterized gravity wave sources; this distinction is described in more detail below.

Properties of the models (Tables 6 and 7) that are of particular relevance for simulating a QBO are:

-   **Vertical domain and resolution:** A high upper boundary is potentially important depending on how much influence the
semi-annual oscillation has on the timing of the start of each new descending QBO cycle. Likewise vertical resolution is important both for accurately simulating vertically propagating equatorial waves and for representing the wave dissipation and descending sharp shear zones that are a characteristic feature of the QBO. Figure 4 (see also columns 3 and 4 of Table 6) shows the different vertical resolutions used by the QBOi participating models (although note that a small number of models share common vertical grids), along with the vertical resolution of the set of 30 diagnostic pressure
levels described in Section 4.1.

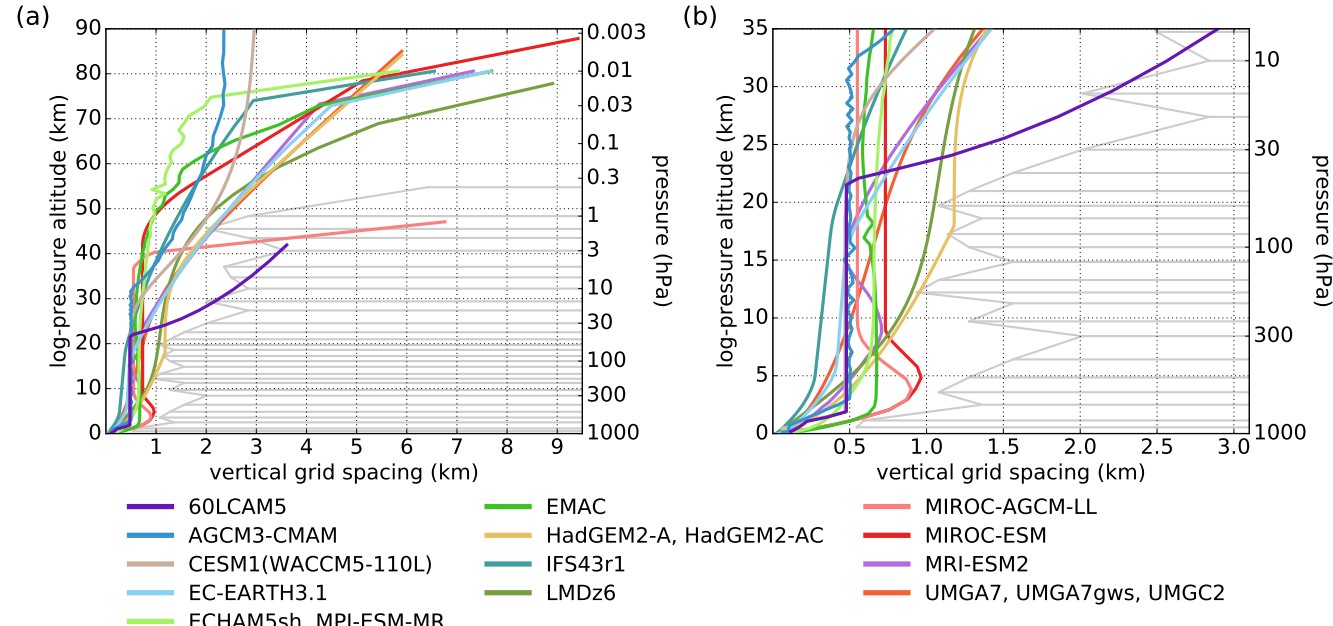

**Figure 4.** (a) Vertical profiles of vertical grid spacing, $\Delta z$ (km), for models participating in QBOi. Log-pressure altitude on the model levels is calculated by assuming a surface pressure of 1013.25 hPa and fixed scale height of 7 km. The grey *horizontal* lines denote the set of 30 QBOi diagnostic pressure levels (see Section 4.1) while the grey *vertical profile* (left end of the grey horizontal lines) indicates the $\Delta z$ of the diagnostics. (b) Same as in (a), but zoomed in to the altitude range most relevant for the QBO.



– **Horizontal resolution:** This is likely to have a significant impact on the development and evolution of wave sources in the tropical troposphere, which are important for forcing the QBO. Horizontal resolution may also affect the propagation and breaking of large-scale Rossby waves propagating from the extratropics, which are now known to affect the QBO (e.g., Osprey et al., 2016). Figure 5 (see also column 2 of Table 6) shows the horizontal resolution of each model and how the differences in horizontal resolution compare to the differences in stratospheric vertical resolution.

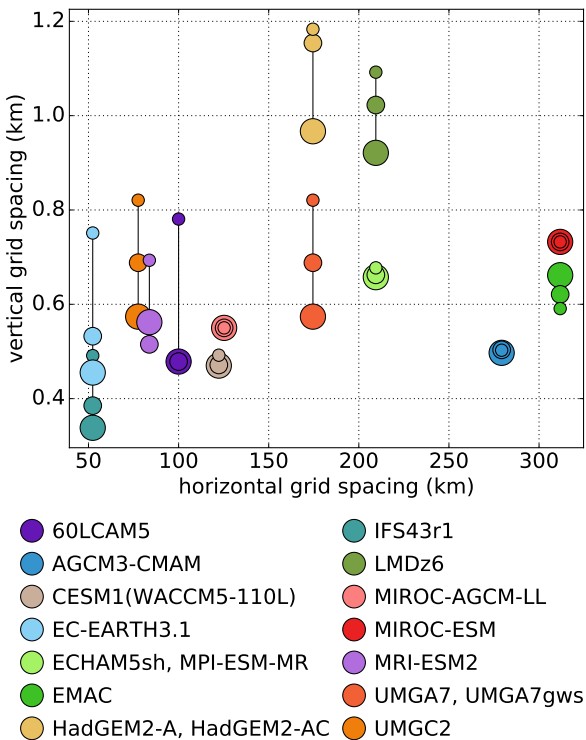

**Figure 5.** Vertical resolution, $\Delta z$, vs. horizontal resolution of models participating in QBOi. Since $\Delta z$ can vary with altitude, as shown in Figure 4, here $\Delta z$ is shown for the three layers 10-15, 15-20 and 20-25 km spanning the tropical upper troposphere and lower stratosphere, with the size of the markers scaled by the average density in each layer. See column 3 of Table 6 for total number of vertical levels in each model. The horizontal grid spacing is estimated by calculating the average of the zonal and meridional grid spacings, $(\Delta\lambda + \Delta\phi)/2$, and converting this to a value in km at the equator. For spectral models with triangular truncation we assume $\Delta\lambda = \Delta\phi = \frac{2}{3}180°/(T+1)$ as an estimate of the transform grid resolution, where $T$ is the truncation wavenumber as given in column 2 of Table 6.

– **Timestep:** The increasing use of inherently stable advection schemes such as semi-implicit semi-Lagrangian methods allows for longer timesteps than are possible, say, with a more traditional Eulerian advection. While this can lead to significant savings in computing requirements, particularly at higher spatial resolution, an adverse effect is the filtering or damping of high frequency equatorial waves (e.g., Shutts and Vosper, 2011) that can potentially make a significant



contribution to the QBO momentum budget. See column 5 of Table 6 for the different dynamical timesteps used by the participating models.

– **Numerical advection scheme:** Model dependence of the QBO on numerical advection schemes generally arises through a sensitivity of the wave propagation characteristics and, perhaps more importantly, the strength of the Brewer-Dobson circulation (Butchart, 2014) due to the tropical upwelling which opposes the descending QBO cycles in the standard paradigm (Baldwin et al., 2001).

– **Parameterized sub grid-scale waves (non-orographic gravity waves):** A very significant development in models that has led to increased success in simulating QBO-like variability has been the introduction of non-orographic GWD parameterizations. Early schemes focused on parameterizing the (vertical) propagation and dissipation of sub grid-scale waves from spatially and temporally fixed sources while more recent developments have included parameterized sources too (e.g., Beres et al., 2005; Choi and Chun, 2011; Lott and Guez, 2013; Schirber et al., 2014; Bushell et al., 2015). Broadly speaking there have been two approaches to parameterizing the propagation and dissipation. The first, followed by Hines (1997a, b) and Warner and McIntyre (1996), aims to represent a broad spectrum of unresolved gravity waves generated by a variety of sources, while the alternative method is to represent the wave spectrum by a finite number, or collection of monochromatic waves such as described by Lindzen (1981) or Alexander and Dunkerton (1999). All models or model-versions participating in QBOi, with the exception of MIROC-AGCM-LL, include at least one parameterization of non-orographic GWD, with the superscripts † or ‡ in the second column of Table 7 indicating, respectively, whether the spectrum or collection of monochromatic waves method is used. A comparison of how the different schemes attenuate parameterized eastward and westward momentum fluxes of non-orographic gravity waves propagating upward through typical wind profiles with opposite phases of the QBO is shown in Figure 7 and described in detail in Section 5.1, below.

Five of the 17 models [60LCAM5, CESM1(WACCM5-110L) HadGEM2-AC, LMDz6 and UMGA7gws] have extended their non-orographic GWD parameterizations to include parameterized gravity wave sources[1]. References giving details of these extended parameterizations are listed in column 3 of Table 7. In most cases this has simply involved replacing an ersatz "fixed" source with one that is more physically based, although for the LMDz6 model the previously-used Hines scheme was replaced with a new GWD parameterization (Lott et al., 2012; Lott and Guez, 2013). There are two pairs of models that are identical except for their gravity wave source being fixed / parameterized: UMGA7 / UMGA7gws and HadGEM2-A / HadGEM-AC. Hence it will be possible to assess the impact these model developments have on the simulation of the QBO and how it responds to changes in climate forcings, at least for a small subset of the participating models.

– **Convection:** An important source of equatorial waves in the models is convection and its associated diabatic heating. Gravity wave source parameterizations also typically couple the generation of parameterized GWD to parameters

---

[1] A "source parameterization" denotes a gravity wave source that is coupled with other physical fields in the model, such as precipitation or deep convective heating, and therefore varies temporally and spatially. In contrast, "fixed" gravity wave sources are not coupled to other physical fields. Fixed sources are often constant in time, although this category could also include sources that have a prescribed temporal variation (e.g. seasonal cycle) or are stochastic.




obtained from the convection schemes such as the precipitation (e.g., Lott and Guez, 2013). The different convection schemes used by the participating models are listed in column 4 of Table 7 for easy comparison.

– **Ozone climatology and feedbacks when interactive chemistry is included:** Although differences in ozone climatologies can potentially impact on simulated QBOs (e.g., Bushell et al., 2010), precise specifications for the ozone forcing were not included in the experiment descriptions (Section 3; Appendix A) to allow for the inclusion of models with prognostic ozone and also to keep the experiment specifications as simple as possible. Therefore for those models without ozone chemistry there are some variations among the ozone climatologies that have been prescribed. Figure 6 illustrates these variations in the tropics, for the ozone used in the timeslice experiments (Experiments 2–4).

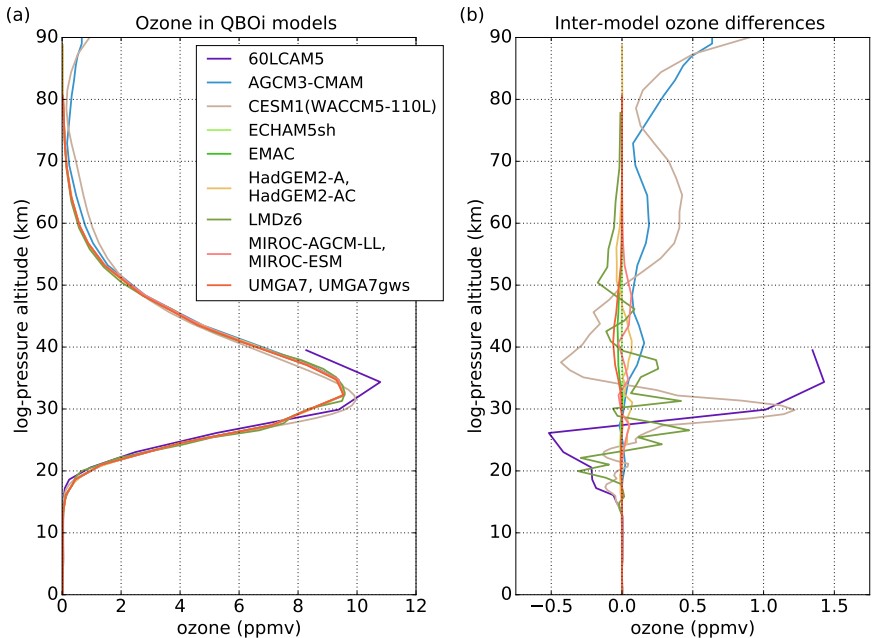

**Figure 6.** (a) Vertical profiles in the tropics of the ozone concentration prescribed in QBOi timeslice experiments (Experiments 2–4; Table 5 indicates which models have performed these experiments), for models that do not include ozone chemistry (as indicated in Table 7). Each vertical profile is an average over the 5°S–5°N latitude band, zonal mean, and annual mean. (b) As (a), but showing differences from a reference profile so that inter-model variations are more clearly visible. The reference profile is the 1988-2007 climatology of the SPARC ozone referred to in Appendix A (item A1; under item A2, 1988-2007 is the SST and sea ice climatological period recommended for Experiment 2).

## 5.1 Offline comparison of non-orographic gravity wave drag schemes

As noted above, non-orographic GWD parameterizations have been important for the generation of a QBO in many climate models (as Table 7 indicates, only one of the QBOi models does not use parameterized GWD). The non-orographic GWD





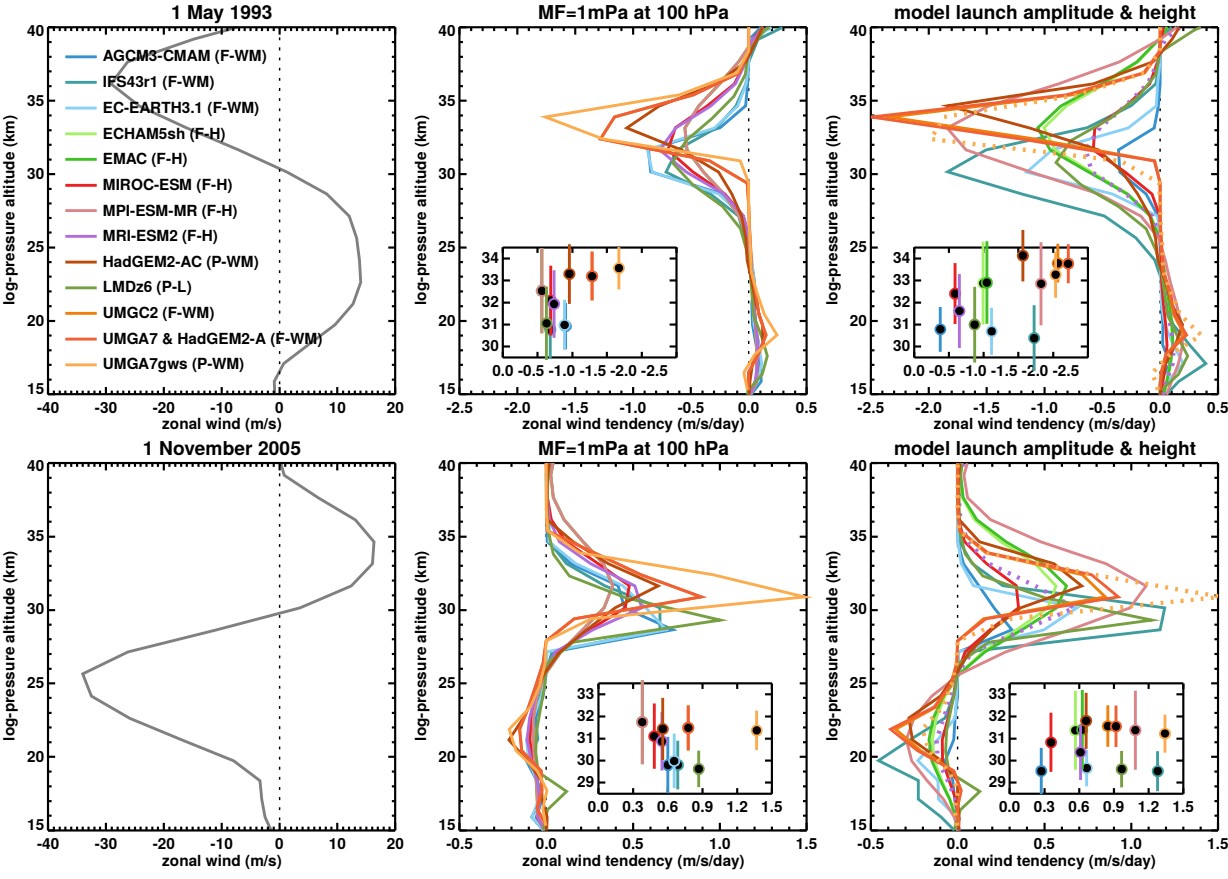

**Figure 7.** Vertical profiles of zonal-mean non-orographic GWD computed using the parameterization schemes used by the different models. The offline calculations are performed using ERA-Interim equatorial zonal and meridional winds and temperatures for $1^{st}$ May, 1993 (top) and $1^{st}$ November, 2005 (bottom). The middle panels show results for the case where the momentum flux is set to $1\,\mathrm{mPa}$ at $100\,\mathrm{hPa}$ ($\approx 16\,\mathrm{km}$). The right panels show results for the case where the models' own launch amplitudes and launch heights are used. Note that the results in the right-hand panel for MRI-ESM2 and UMGA7gws have been multiplied by 0.1 and 0.6, respectively, and the GWD profiles plotted using dotted lines (see Appendix B). The labels in parentheses to the right of the model names denote the type of GWD scheme: "F" or "P" for fixed or parameterized sources; "H" for Hines, "WM" for Warner-McIntyre, or "L" for Lott et al. (2012) for the type of dissipation used. Note that "WM" here includes both the Warner and McIntyre (1999) and Scinocca (2003) schemes (Table 7), which are both implementations of the Warner and McIntyre (1996) framework for gravity wave parameterization. The insets show the parameters of Gaussian fits, $A \exp[-((z-B)/C)^2]$, to the zonal-mean GWD profiles. The peaks of the Gaussians ($A$, $\mathrm{m\,s^{-1}}$ per day, horizontal axes) and their heights ($B$, km, vertical axes) are denoted by the filled circles. The $e$–folding widths of the Gaussians ($C$, km) are given by the vertical bars. See text and Appendix B for more details.

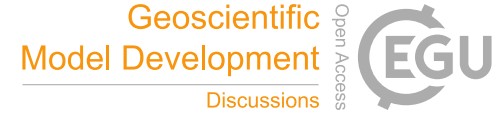



parameterization schemes used in the QBOi models are compared by performing offline calculations for prescribed equatorial wind and temperature profiles (see Appendix B for details). The $1^{st}$ May, 1993 and $1^{st}$ November, 2005 start dates for Experiment 5 (Figure 3) are used since they have oppositely phased QBOs. Three experiments are performed. The first two use a prescribed amount of momentum flux (MF) at a launch height[2] of 100 hPa, namely 1 mPa and 10 mPa. Inter-model differ-

ences in GWD in these two experiments arise solely from differences in the phase speed spectrum at the launch height and the nonlinear dissipation mechanism inherent in the schemes (e.g., Hines' Doppler spreading or Warner and McIntyre's imposed saturated spectrum). The purpose of the 10 mPa experiment is to see how linearly the MF (and GWD) scales with the MF at 100 hPa in comparison with the 1 mPa experiment. The third experiment uses the models' own launch heights and amplitudes; hence this experiment most closely matches the setup used in the QBOi simulations. For all three experiments the GWD is

computed at each longitude and the results zonally averaged.

Vertical profiles of zonal-mean GWD for the 1 mPa experiment are shown in the middle panels of Figure 7. Results for the 10 mPa experiment (not shown) are quite similar to the 1 mPa results but are larger by a factor of ten, confirming that to a good first approximation the GWD at these heights scales linearly with the MF at 100 hPa. This is perhaps not too surprising given that critical level absorption by the background winds, as opposed to nonlinear dissipation resulting from the exponential

growth with height of the gravity wave amplitudes, is the primary cause of the momentum flux deposition in these highly sheared wind profiles. The results of the third experiment are shown in the right panels. Compared to the 1 mPa results, these show much more inter-model spread. Since the source specifications used in this experiment are the ones that produce each model's best QBO, the larger inter-model spread in the third experiment is a reflection of model dependent biases in, for instance, the mean winds and temperatures, and resolved waves that must be overcome by tuning of the gravity wave sources.

The GWD profiles between 20 and 40 km are approximately Gaussian in form and can be simplified by fitting the zonal-mean GWD to a function of the form $A \exp[-((z-B)/C)^2]$. The three fit parameters are shown in the insets in the middle and right panels. The increase in inter-model spread of the maximum GWD (fit parameter A) in the experiment using the models' launch amplitudes and heights is more readily seen. As observed (not simulated) precipitation is used in the offline calculations for two of the models using parameterized gravity wave sources (LMDz6 and UMGA7gws), the results in the right-hand panels

of Figure 7 may not accurately reflect what the models themselves would produce. Hence the parameterized-source and fixed-source results in the right-hand panels are not entirely comparable. A case in point is the rather large difference in the peak GWD in the UMGA7 (fixed source) and UMGA7gws (parameterized source) results; for this reason the UMGa7gws results have been scaled to fit on the plot. Note also in the 1 mPa experiment that the GWD peaks are wider in the vertical and weaker for the models that use Hines than for the others. This is consistent with the vertical smoothing of the momentum fluxes that

is conventionally applied in the Hines scheme before the GWD is computed. The differences in the 1 mPa Hines results are a consequence of the different amount of smoothing used by the different models; if the smoothing is removed from the offline calculation, the 1 mPa Hines results for the different models are identical.

---

[2]For the models with parameterized gravity wave sources, this "launch height" is instead a reference height at which the offline scheme are tuned to have the specified properties; see Appendix B for further details.





In summary, the offline comparison shows that most of the inter-model differences in the parameterized GWD in the equatorial stratosphere arise from the differences in their launch height and launch amplitude, not from differences in the wave dissipation mechanism and the shape of the assumed launch spectrum.

## 6   Closing remarks and future plans

The QBO is arguably the most conspicuous and regular mode of variability observed anywhere in the atmosphere that is not directly related to either the annual or diurnal cycles. At a fundamental level, and for current conditions, it can be considered to be purely an atmospheric dynamical mode of variability, despite possible external influences from variability in the oceans, the solar cycle or changes in atmospheric composition. Therefore the primary goals of phase 1 of QBOi are achievable using atmosphere-only global models that are computationally relatively inexpensive to run. To date (July 2017) output from 17 models/model versions (Table 5) has been uploaded, or is planned for uploading, to the shared database.

The goals of phase 1 of QBOi are to:

– Compare, for present day conditions, the accuracy of the morphology of the simulated QBOs across models, and relate this to differences between models in the representation of the forcing mechanisms (e.g., terms contributing to the zonal-mean zonal momentum equation) and other model properties such as resolution and sources of waves.

– Compare how the morphology of the simulated QBOs and QBO forcing mechanisms respond to climate change (i.e., a doubling and quadrupling of $CO_2$ amounts) and identify which aspects of these responses are robust.

– Compare QBO predictive skill between models and its dependence on the QBO's initialized phase, the underlying state of the atmosphere and/or properties of the individual models (e.g., why was there an absence of skill in predicting the disruption of the QBO in 2016?).

Phase 1 of QBOi therefore addresses the challenges associated with modelling, predicting the evolution of, and projecting long term changes in the QBO. Results from planned studies are expected to inform on requirements for future model development leading to more accurate representations of the QBO and its variability in the individual models and across the multi-model ensemble. Benefits, however, are likely to extend well beyond this and range from potential enhancements in skill in seasonal to decadal predictions resulting from concomitant improvements in QBO-extratropical dynamical teleconnections, to better capabilities for assessing the consequences of geoengineering proposals involving the injection of aerosol into the equatorial stratosphere where its redistribution away from the tropics is likely to be significantly influenced by the QBO.

Beyond phase 1, QBOi is expected to focus more on QBO extratropical dynamical teleconnections and couplings to other aspects of the climate system. In this respect QBOi again differs from those multi-model activities like CMIP and CCMI that are largely policy-driven and hence place considerable emphasis on continually updating projections using the latest generation of models. Instead the developing consensus in the QBOi community, which has emerged primarily from the September 2016 QBO workshop (see Anstey et al., 2017, for a workshop summary), is to build on the experiments described in this paper though, of course, results from phase 1 studies are expected to feed through into improving the representation of the QBO





in the next generation of models. Some new coordinated studies that have been proposed for future endorsement by QBOi include:

- Increasing the ensemble size of Experiment 1 ("AMIP") to examine the robustness across models of possible synchronisation bewteen ENSO events and the QBO (e.g., Christiansen et al., 2016).

– Extending Experiment 2 (present-day time slice) to increase the sample size to examine QBO teleconnection robustness in an idealised framework in which there is no other externally forced variability, apart from the annual and diurnal cycles.

- Repeating Experiment 2 (present-day time slice) with idealised perpetual El Niña / La Niña SST anomalies to examine the interaction of ENSO and QBO teleconnections.

– Empirically separating the effects of stratospheric and tropospheric climate change on the QBO by modifying Experiments 3 and 4 (future time slice) such that the increases in $CO_2$ amount (~forcings stratospheric climate change only) and SSTs (~forcing tropospheric climate change only) are applied separately.

- Extending Experiment 5/5A (retrospective hindcasts) to examine the 2016 QBO disruption and its predictability.

- Examining the impact of ozone on the QBO either through prescribed ozone perturbations or through ozone feedbacks
15       for those models that can rerun with and without ozone chemistry.

The above list is by no means exhaustive and other possible extensions of the research plans for QBOi include more idealized studies comparing simulations using only "dynamical cores" (e.g., Yao and Jablonowski, 2015) or perhaps simulations in which the QBO is artificially removed (e.g., by turning off the non-orographic GWD parameterization in the tropics). However, in line with current QBOi practices, details of any new coordinated studies will again be formulated through community discussion
at forthcoming QBOi workshops, and will depend on the outcomes of the phase 1 studies.

*Code and data availability.* For information on the code availability for the individual models considered in this paper see the appropriate references given in Table 5. Details of the QBOi data repository and how to access it are provided in the Supplementary.

## Appendix A: Experiments - technical specifications

### A1    Experiment 1 - "AMIP"

Experiment 1 is based on the CMIP5 Expt 3.3 alternatively referred to as the "Atmospheric Model Intercomparison Project (AMIP)" experiment (Taylor et al., 2012). It is a 1-3 member ensemble of 30-year simulations using observed SSTs and sea ice amounts from $1^{st}$ January 1979 to $28^{th}$ February 2009. These can be obtained from:



http://www-pcmdi.llnl.gov/projects/amip/AMIP2EXPDSN/BCS/amipbc_dwnld.php

The corresponding external forcings for the CMIP5 AMIP-experiment (e.g., radiative trace gas concentrations, aerosol distributions, solar irradiance, and appropriate forcings from explosive volcanoes) can be found here:

http://cmip-pcmdi.llnl.gov/cmip5/forcing.html#amip

apart from ozone which, for high-top models, can be obtained from:

https://groups.physics.ox.ac.uk/climate/osprey/QBOi_O3/

Initial conditions are not prescribed and it is left to individual groups to use whatever is appropriate for their model and to include any spin-up if this is considered necessary.

## A2 Experiment 2 - $1\times CO_2$

Experiment 2 is similar to Experiment 1 but with a repeated annual cycle for the SSTs and sea ice amounts plus all the other forcings (i.e., there is no interannual variability or any secular changes in the forcings). It can either be a 1-3 member ensemble of 30-year simulations or *preferably* a single 100-year (or longer) simulation. The long single integration has the additional potential of providing information on very low frequency variations.

Ideally the external annual cycle forcings should be 30-year climatologies based on Experiment 1 although, as these are
generally not readily available, a suitable alternative is to apply annually repeating forcings based on the 2002 CMIP5 forcings. The year 2002 is well removed from any explosive volcanic eruptions and the ENSO and Pacific Decadal Oscillation (PDO) are both in their neutral phases and hence conditions is this year can be considered as a useful proxy for the multi-year mean for most quantities. However 2002 ozone amounts are likely to be strongly perturbed because of the Southern Hemisphere sudden stratospheric warming (e.g., Shepherd et al., 2005) and for ozone a 2D climatological field representative of the 1990s
is preferable. For SSTs and sea ice amounts CMIP5 1988-2007 climatologies are available from:

http://www-pcmdi.llnl.gov/projects/amip/AMIP2EXPDSN/BCS/amipbc_dwnld.php

As Experiment 2 is the control for Experiments 3 and 4 ($2\times CO_2$ and $4\times CO_2$, respectively) the average $CO_2$ amount for 2002 should be used as the baseline $1\times CO_2$ amount.

Although the use of different length climatologies for different forcings is not ideal and does not provide direct comparison
to the 30-year period of Experiment 1, the observed dependence of the QBO on a changing climate through this period appears to be negligible. Thus for QBOi the benefits of the simpler experimental set-up is considered to far outweigh any possible disadvantages. Nonetheless it important to emphasize that the same idealised set of climatologies and forcings are to be used throughout Experiments 2-4, that is, apart from the changes to the $CO_2$ amounts and SSTs described below.

As with Experiment 1, atmospheric initial conditions are not prescribed.

## A3 Experiments 3 - $2\times CO_2$, and 4 - $4\times CO_2$

Experiments 3 and 4 are the same as Experiment 1 but for "$2\times CO_2$" and "$4\times CO_2$" climates, respectively. Again these can either be a 1-3 member ensemble of 30-year simulations, or *preferably* a single 100-year simulation, after allowing for a suitable spin-up to the new climate (without a coupled ocean this is expected to be fairly rapid though for the $4\times CO_2$ experiment this can

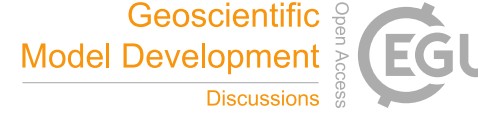



be of order five years). Compared to the amount specified for Experiment 1 the $CO_2$ concentration should be either doubled (Experiment 3) or quadrupled (Experiment 4) with a corresponding idealized adjustment made to the SSTs of a spatially uniform perturbation of +2K for 2×$CO_2$ and +4K for 4×$CO_2$. Sea ice amounts should be kept the same as in Experiment 1.

All other forcings in these two Experiments should be *exactly* the same as in Experiment 1 including the amounts of all
radiatively active greenhouse gases other than $CO_2$. If ozone is prescribed (i.e., if the model does not have interactive chemistry) then this too should be exactly the same as in Experiment 1. Alternatively if the model does have interactive chemistry then the source gases and/or emissions should be kept exactly the same as in Experiment 1. This idealized set-up for Experiments 3 and 4 is appropriate as these are sensitivity experiments and not attempts to predict specific periods in the future.

As with Experiment 1 atmospheric initial conditions are not prescribed, but note the need to allow for spin-up to the new
climates.

## A4   Experiment 5 - QBO hindcasts

These are atmosphere-only experiments, initialized from reanalysis data, providing multiple ensembles of short integrations from a relatively large set of start dates sampling different phases of the QBO. The prescribed start dates (i.e., atmospheric initial conditions) are $1^{st}$ May and $1^{st}$ November for the years 1993-2007 (i.e., 15 years with a total 30 start dates). The
duration of each hindcast should be at least 6 months but preferably 9-12 months.

As with Experiment 1 the boundary conditions and external forcings should be the same as those specified for the CMIP5 AMIP experiment (Taylor et al., 2012). CMIP5 interannually varying sea ice and SSTs can be obtained from:

http://www-pcmdi.llnl.gov/projects/amip/AMIP2EXPDSN/BCS/amipbc_dwnld.php

while the CMIP5 external forcings for radiative trace gas concentrations, aerosols, solar, explosive volcanoes, etc., can be
obtained from:

http://cmip-pcmdi.llnl.gov/cmip5/forcing.html#amip

Ozone forcing datasets appropriate for use in high-top models are available from:

https://groups.physics.ox.ac.uk/climate/osprey/QBOi_O3/

Initial data for the hindcasts should be taken from the ERA-Interim reanalysis (Dee et al., 2011) which can be downloaded
from:

http://apps.ecmwf.int/datasets

Registration is required; if downloading many start dates from this site, it may be easier to use the "batch access" method described on the site, although interactive download of each date is also possible. Data are available on either standard pressure levels or original model levels, and in either grib or netCDF formats. The ensemble is expected to be generated by perturbing
the initial conditions by a small anomaly, which needs do no more than change the bit pattern of the simulation. For some models this is possible through stochastic physics, however each group should use an ensemble generation method that is most appropriate to their model and that is most readily available to them.



## A5 Experiment 5A - QBO forecasts

This experiment is as Experiment 5, but using a coupled ocean-atmosphere model and predicting the SST, instead of specifying observed values. External forcings should also be fixed at the initial start time so as not to use future information. This is then a true forecast experiment for the QBO, and can be compared with the results of Experiment 5. Some groups may already have
performed these hindcasts as part of their operational seasonal forecasts but note that for QBOi purposes it is important that the majority of the diagnostics discussed in Section 4 are available for a full comparison to Experiment 5 results.

## Appendix B:  Offline non-orographic gravity wave drag calculations

This appendix provides details about the offline GWD calculations shown in Figure 7. The background equatorial winds and temperatures are from a single day (daily mean) of ERA-Interim data on a $1°$ longitude grid and on pressure levels at the
ECMWF model levels resolution.

For models that use "fixed" gravity wave sources (e.g., AGCM3-CMAM), the calculations are straightforward and simply involve computing the GWD above the launch height. Since these models all use a horizontally isotropic gravity wave source, the MF in a single azimuth is set to either 1 or 10 mPa for the first two experiments. All fixed-source calculations are done using offline versions of the Scinocca  (2003), Hines (1997a, b) and Warner and McIntyre (1999) non-orographic GWD schemes
using each model's parameter settings. Results for the third offline experiment in which the models' own source amplitudes (i.e., momentum flux for Scinocca, root-mean-square (RMS) winds for Hines) and launch heights are used, are validated by comparing to results from QBOi Experiment 5 for models that provided daily-mean GWD. With the exception of one model, the agreement is reasonably good which is all that can be expected given that the resolution of the models differs from that used in the offline calculations. For MRI-ESM2 the offline results for the third experiment are ten times larger than the Experiment
5 results, and have been scaled in the right panels of Figure 7. The reason for this large discrepancy is unknown. For models that tie their non-orographic gravity wave sources to parameterized processes in the troposphere (referred to in the Figure 7 caption as parameterized sources), the calculations are more involved.

For the models that were able to perform the offline calculations for parameterized-source schemes (LMDz6, UMGA7gws and HadGEM2-AC) the procedure was as follows. For LMDz6, daily precipitation observations were used to generate an
ensemble of monochromatic waves. The background winds and temperatures are held fixed in time using either the $1^{st}$ May or $1^{st}$ November data. A similar procedure is used for the other two models, except that the launch momentum fluxes in HadGEM2-AC are obtained by sampling from the Experiment 1 result for the month since the source parameterization in HadGEM2-AC requires convective heating profiles not provided by observations. As momentum flux is not prescribed for these models, tuning of the gravity wave parameters is required to achieve the desired MF at 100 hPa for the first two experiments,
such that $(|MF_{east}|+|MF_{west}|)/2 = 1$ or 10 mPa at 100 hPa. Due to time constraints the NCAR group, which also ties its GWD scheme to convection in the 60LCAM5 and CESM1(WACCM-L110) models, was unable to participate in this comparison.





*Acknowledgements.* The design of the experiments described here grew out of community discussions at the first QBOi workshop in March 2015 in Victoria, Canada. Funding for the workshop from the UK Natural Environment Research Council (NE/M005828/1), the World Climate Research Programme (WCRP), Stratosphere-troposphere Processes and their Role in Climate (SPARC) activity and the Canadian Centre for Climate Modelling and Analysis is gratefully acknowledged. We further acknowledge the scientific guidance of the WCRP for

helping motivate this work, coordinated under the framework of the SPARC QBO initiative (QBOi) led by JA, NB, KH and SO. The Centre for Environmental Data Analysis (CEDA) have very kindly offered to host the QBOi data archive. NB and AS were supported by the Joint UK BEIS/Defra Met Office Hadley Centre Climate Programme (GA01101). SO was supported by NERC projects NE/M005828/1 and NE/P006779/1. SW and YK used the Earth Simulator for QBOi simulations and were supported by the SOUSEI program, MEXT, Japan and Japan Science and Technology Agency (JST) as part of the Belmont Forum. YK was supported by Grant-in-Aid for Scientific

Research B (26287117) and Joint international Research (15KK0178) from the Japan Society for the Promotion of Science, and by the Environment Research and Technology Development Fund (2-1503) of the Ministry of the Environment, Japan. FL and SO were supported by the ANR/JPI-Climate/Belmont Forum project GOTHAM (ANR-15-JCLI-0004-01). FS was supported by the European Commission, under grant number StratoClim-603557-FP7-ENV.2013.6.1-2, with computing resources for the ECHAM5sh simulations provided by an ECMWF Special Project. YHK was supported by Basic Science Research Program through the National Research Foundation of Korea funded by

the Ministry of Science, ICT & Future Planning (NRF-2015R1C1A1A02036449). HP was supported by the German Federal Ministry for Education and Research (BMBF) project MiKlip (FKZ 01LP1519A) and thanks Elisa Manzini for providing additional information on the MPI model. BSC contribution is supported by the Spanish MINECO-funded DANAE project (CGL2015-68342-R) and EU H2020-funded MSCA-IF-EF DPETNA project (GA No. 655339).





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
