# Peer review of "Overview of experiment design and comparison of models participating in phase 1 of the SPARC Quasi-Biennial Oscillation initiative (QBOi)"

_Geoscientific Model Development, 2017_

## Referee Comment (RC1) · Anonymous Referee #1 · 16 Nov 2017

SUMMARY: The Quasi-Biennial Oscillation (QBO) is the largest mode of tropical wind variability and, with a period of approximately 28 months, one of the largest regions of interannual variability in the atmosphere. Its basic mechanism has long been understood, yet a realistic representation of its behavior, structure, along with accurate seasonal QBO forecasts are often difficult to realize in global atmospheric models. These limitations are being addressed by the SPARC QBOi program. This paper describes the QBOi planned experiments and participating models, along with descriptions of requested output and targeted diagnostics. In addition, this paper includes comparisons of participating gravity wave drag parameterizations (GWDP), a key driver of the modeled QBO circulations. The goals, experiments, and relevant model parameters are

well documented. The figures are excellent and appropriate to the discussion. The abstract and tables are clear and complete. Results from the GWDP comparisons show that characterizing the wave sources (launch altitudes and wave amplitudes) is more important than the details of the parametrized wave propagation and breaking, at least in obtaining consistent results across models. Overall this is a well written documentation of a significant modeling community effort to improve global models for climate and seasonal forecasting.

STRENGTHS: This paper highlights the many factors that can influence the QBO, from model numerical advection schemes to the important GWD by both resolved and parameterized waves. It forms a crucial baseline reference for further QBOi publications and should be a useful reference for QBO studies outside of the QBOi program.

WEAKNESSES: No major weaknesses. A few minor comments are listed below. The third key question concerning the QBO and CO2 could be rewritten to be more focussed. A couple of additional references are provided in the comments below that should probably be included.

RECOMMENDATION: This manuscript can be published with very minor revisions as suggested below. It should include the two additional references listed below and clarify the third key question.

MINOR COMMENTS:

Page 4, Line 5,: Are there any anticipated difficulties with closing offline momentum budget studies? Maybe that could be discussed here or in the Diagnostics section.

Page 4, Lines 27-29: This section provides a fairly comprehensive list of models with QBO-like variability. The current NASA GEOS model has QBO-like variability (Molod et al., 2015) and should be included as well. Reference: Molod, A., Takacs, L., Suarez, M., and Bacmeister, J.: Development of the GEOS-5 atmospheric general circulation model: evolution from MERRA to MERRA2, Geosci. Model Dev., 8, 1339-1356,

https://doi.org/10.5194/gmd-8-1339-2015, 2015.

Page 7, Lines 13-14: In this third key question about CO2 and climate, does "present-day climate" refer to changes in present climate from recent CO2 changes? Maybe this question needs to be further explained or more focused.

Page 9, Line 1: This section (Process Studies) outlines using the hindcasts to examine wave dissipation and momentum deposition in detail. While mentioned in the future plans in section 6, including an initial time in the spring of 2016 as the anomalous equatorial wind profiles occurring just after the disruption winter of 2015-16 would provide an interesting challenge to both the parameterized and resolved waves.

Page 18,: Lines 3-6: The importance of numerical advection schemes is noted but the specific schemes used are not given for each model. Also, could more detail be given about the relation between the BD-Circulation and specific advection schemes? Do certain schemes consistently give a faster BD circulation?

Page 19, Line 3: The title of the subsection includes the phase "...feedbacks when interactive chemistry is included", however the subsection mostly discusses the ozone climatology for use without interactive chemistry. Maybe the title should be modified or more discussion of potential ozone feedback added.

Page 19, Section 5.1: This section discusses only a subset (13) of the QBOi models. While mentioned in Appendix B, maybe a sentence could be added here to note this restriction. In particular, are the results likely to be different for the Lindzen schemes? Also, I think this should be Section "5" rather than "5.1".

Page 23, Line 3: "ENSO"; This might be a good place to reference a noted connection between the QBO disruption and ENSO: Barton, C. A., & McCormack, J. P. (2017). Origin of the 2016 QBO disruption and its relationship to extreme El Niño events. Geophysical Research Letters, 44. https://doi.org/10.1002/2017GL075576

VERY MINOR COMMENTS

The following references in the reference section do not appear to be in the text:

Page 30, Line 24: Giorgetta et al., 2013

Page 30, Line 35: Hazeleger et al., 2012

Page 34, Line 1: Stevens et al., 2013

Page 34, Line 21: Warner and McIntyre, 2001

Page 30, Lines 15-19: The "a" and "b" are missing from the two Gerber et al. 2016 references.

Page 31, Line 24: The "a" is missing form the Kim et al. 2015 reference.

Page 33, Line 28: Serva et al., is misplaced in the reference list.

---

## Referee Comment (RC2) · Anonymous Referee #2 · 21 Nov 2017

Summary: The paper lies out the rationale for QBOi, defines several experiments that will allow scientific questions about the QBO to be answered, and gives some detailed information on the participating models. Not being an outright expert on the QBO, my view is that the experiments have been well thought out and that the paper will form an adequate basis for QBOi. Weaknesses of the paper include that in a few places unnecessary options are given to the participants that may complicate the evaluation of the results. Also some forcings could be specified in more detail, in order to reduce ambiguity. Furthermore, I am not sure enough relevant detail is conveyed about the internal make-up of the models to allow analysts to understand why models behave the way they do. Maybe an additional appendix could be written, consisting of one-

paragraph descriptions of the individual models, highlighting particular aspects of their make-ups that the PIs consider to be relevant for their simulation of the QBO, that go beyond just their representation of parameterized GWD. Finally, in some places I was unclear as to the function of this paper: The title suggests that the paper is about defining the experiments and describing the models, but in places the text talks about the performances of the models, which I consider out of scope for this paper. Such a discussion of model performance should be reserved for subsequent papers that conduct the scientific analysis.

I recommend publication of the paper subject to answering these comments and also considering the below minor issues:

P1L2: Capitalize "General Circulation and Earth System Models".

P1L4: "Verisimilitude" is an unusual word in this context. While I appreciate stylistic richness, perhaps consider replacing with something more widely understood, such as "realism".

P2L9: has -> have

P2L24: has -> have

Figure 1: Perhaps a sentence can be written about the Unified Model family. HadGEM2-CC produces a realistic QBO, HadGEM2-AO, HadGEM2-ES, and the ACCESS models do not. What is it about HadGEM2-CC that produces this difference in behaviour? Higher lid, more levels in the stratosphere?

P4L31: Cut out "fragile – which is to say"

P5L18: Insert "," after "experiments".

P6L4: You say elsewhere that a 100-year simulation would be preferred. I suggest to remove the options of 3x30 years here and elsewhere. Otherwise you just complicate the analysis and might introduce problems with ensuring that the 3 ensemble members

are sufficiently independent of each other.

Figure 2: I suggest not to fill in the area under the curves in red, blue and black. These areas are overlapping, impossible to see in places and difficult to distinguish. One way to simplify this is to recognize that black is the mean annual cycle (i.e. it's periodic). If you subtract this off the other two signals and display the difference, the figure may become easier to read.

P7L4: For the purpose of clarity, perhaps spell out explicitly what the 1xCO2, 2xCO2, and 4xCO2 mixing ratios actually are (e.g. 300 ppmv, 600 ppmv, 1200 ppmv, or whatever the numbers should be). Also what are the mixing ratios for other GHGs (CH4, N2O, etc.) for these experiments?

P7L19: Insert "some" before "models".

P7L25: I would replace "9-12 months" with 12 months. Otherwise you may get an unnecessary diversity of responses there. Also spell out which reanalyses are supposed to be used here.

P8L14: ditto.

P9L22: What is a "vertical resolution equivalent to the model resolution"? Would it not be easier to say that data are requested on model (hybrid-pressure, in most cases) levels? In this case surface pressure and a recipe for the calculation of model level pressure need to be provided, or full pressure fields for hybrid-height models.

P12L6: Experiment 5A is designed for coupled AOGCMs, so the statement is inaccurate, but maybe models without an option to couple to an ocean can just skip this experiment. (Experiment 5A talks about an "appropriate initialization" of the ocean: What is that? Please provide more detail.)

P19L3: I do not understand why the ozone forcing for models without interactive chemistry is not prescribed. This will be an uncontrolled and unnecessary source of inter-model variations, compounded by the lack of any requirements to document any ozone

forcing used. Since almost all models listed will not have interactive ozone, prescribing ozone, and perhaps also conducting a sensitivity experiment using a variant ozone climatology, might be interesting to include. It would be straightforward e.g. to require models without interactive ozone chemistry to use the CMIP6 historical ozone climatology. An option here is to request variant simulation where ozone is varied from CMIP6 to whatever individual groups prefer, to get an idea about the influence of this choice.

P24L12: Again, I suggest to replace the two options with a simple requirement to produce a single 100-year simulation.

P24L23: To remove ambiguity and make everyone's lives easier, the $CO_2$ volume mixing ratio reached in 2002 should be numerically stated here. (How about the other GHGs?)

P25L15: Replace "9-12 months" with "12 months".

P26L9: Why not require the data on ECMWF model levels, and ditto for other fields? This way no information is lost to interpolation.

Figure 7, section 5.1, appendix B: This is moving into the territory of comparing model performance, which I think would be out of scope for this paper. This off-line comparison is actually an interesting scientific exercise itself, but does not quite fit into this paper, the thrust of which is on describing the QBOi experiments. The brevity of presentation afforded to it here also does not quite do it justice. Please consider expanding this into a compact stand-alone publication (which might be submitted to GMD and would be an interesting companion paper to this one).

---

## Referee Comment (RC3) · Anonymous Referee #3 · 24 Nov 2017

Review of "Overview of experiment design and comparison of models participating in phase 1 of the SPARC Quasi-Biennial Oscillation initiative (QBOi)" by Neil Butchart and many others

Recommendation: Minor revisions

This study introduces the model integrations performed as part of the first phase of the QBOi, a model intercomparison project that hopes to shed light on the processes that lead to a spontaneous QBO and how they will change in the future. Such a project is sorely needed in our field, and I look forward to reading future papers that utilize this model output. I have only a few minor comments, and after the authors address them

the paper will be ready for publication.

1. Figure 2: The use of filled black patches for the "mean annual cycle" forcing is visually confusing. I suggest a thick line.

2. It is too late to correct this, but in retrospect there probably should have been guidance for the ozone profile to be used for models without interactive chemistry. There are ozone-temperature feedbacks in this region that will differ among models, and unraveling the causes of these feedbacks will likely be hard. Again, I don't think it is worth rerunning experiments, and hopefully the archived ozone will suffice.

3. The numerical, thermal, and mechanical dissipation used by each model likely differs, and these three sources of dissipation might be important for the QBO momentum budget in some models (e.g. Yao and Jablonowski,2015, already cited). I have two suggestions: first, please ask the models to submit their wind and temperature tendencies due to these three sources of dissipation (or at least the total tendency due to dissipation)! Zonal and monthly mean is probably good enough. Second, please add a column to table 6 or 7 (or a new table) where each model reports on how it implements numerical, thermal, and mechanical dissipation. It might also be helpful for each model to state which advection scheme/dynamical core it uses.

4. Figure 7, top left panel: I suggest writing the model names in color. Also, two colors appear to be used for more than one model (at least to this reviewer's mildly color-blind eye). Specifically, the shade of red used for MIROC-ESM (F-H) and HadGEM2-AC (P-WM) is very similar. Similarly, the shade of green used for LMDz6 (P-L) and EMAC (F-H) is very similar. Please adjust the colors to add more contrast.

5. Page 23, line 4 "between" is misspelled

6. Table 4: it would be helpful if one level near 200hPa was also included, as one might want to compare the upper tropospheric resolved wave spectrum (i.e. Wheeler and Kiladis 1999 diagrams) among models. That is, the resolved wave spectrum near

the top of convection will differ among the models (possibly due to different convection schemes used by each model), and it might be interesting to relate any differences in QBO morphology to differences in tropospheric wave generation that are in turn related to convection schemes. This additional level will also allow one to study the affect of vertical resolution in the TTL on resolved wave vertical propagation – it is conceivable that models with coarser vertical resolution will have stronger degradation in their resolved wave fluxes between $\sim$200hPa and $\sim$100hPa.

7. The native vertical levels of each model (i.e. the data underlying figure 4) should be made accessible, perhaps as a data supplement or hosted on the QBOi website.

---

## Author Comment (AC1) · 12 Jan 2018

SUMMARY: The Quasi-Biennial Oscillation (QBO) is the largest mode of tropical wind variability and, with a period of approximately 28 months, one of the largest regions of interannual variability in the atmosphere. Its basic mechanism has long been understood, yet a realistic representation of its behavior, structure, along with accurate seasonal QBO forecasts are often difficult to realize in global atmospheric models. These limitations are being addressed by the SPARC QBOi program. This paper describes the QBOi planned experiments and participating models, along with descriptions of requested output and targeted diagnostics. In addition, this paper includes comparisons

of participating gravity wave drag parameterizations (GWDP), a key driver of the modeled QBO circulations. The goals, experiments, and relevant model parameters are well documented. The figures are excellent and appropriate to the discussion. The abstract and tables are clear and complete. Results from the GWDP comparisons show that characterizing the wave sources (launch altitudes and wave amplitudes) is more important than the details of the parametrized wave propagation and breaking, at least in obtaining consistent results across models. Overall this is a well written documentation of a significant modeling community effort to improve global models for climate and seasonal forecasting.

STRENGTHS: This paper highlights the many factors that can influence the QBO, from model numerical advection schemes to the important GWD by both resolved and parameterized waves. It forms a crucial baseline reference for further QBOi publications and should be a useful reference for QBO studies outside of the QBOi program.

WEAKNESSES: No major weaknesses. A few minor comments are listed below. The third key question concerning the QBO and CO2 could be rewritten to be more focussed. A couple of additional references are provided in the comments below that should probably be included.

RECOMMENDATION: This manuscript can be published with very minor revisions as suggested below. It should include the two additional references listed below and clarify the third key question.

>The authors thank the reviewer for the rather positive comments. The suggestions for additional references have been >added and the third key question revised.

Page 4, Line 5,: Are there any anticipated difficulties with closing offline momentum budget studies? Maybe that could be discussed here or in the Diagnostics section.

>Some discussion has been added to the Diagnostic section (end of section 4.1).

Page 4, Lines 27-29: This section provides a fairly comprehensive list of models with

QBO-like variability. The current NASA GEOS model has QBO-like variability (Molod et al., 2015) and should be included as well. Reference: Molod, A., Takacs,L., Suarez, M., and Bacmeister, J.: Development of the GEOS-5 atmospheric general circula-tion model: evolution from MERRA to MERRA2, Geosci. Model Dev. 8, 1339-1356, dis-cussion paper: https://doi.org/10.5194/gmd-8-1339-2015, 2015.

>Reference to the NASA GEOS model has been added to make the list more compre-hensive.

Page 7, Lines 13-14: In this third key question about CO2 and climate, does "present-day climate" refer to changes in present climate from recent CO2 changes? Maybe this question needs to be further explained or more focused.

>The third key question has been rephrased.

Page 9, Line 1: This section (Process Studies) outlines using the hindcasts to exam-ine wave dissipation and momentum deposition in detail. While mentioned in the future plans in section 6, including an initial time in the spring of 2016 as the anomalous equa-torial wind profiles occurring just after the disruption winter of 2015-16 would provide an interesting challenge to both the parameterized and resolved waves.

>We agree that 2016 disruption provides a challenge for initialised forecast. However this section is documenting the >plans for phase 1 of QBOi which have already been agreed by the community and in many cases executed. Therefore >no changes have been made to the Process Studies subsection. Nonetheless as a community we do recognise the >new challenges thrown up by the 2016 disruption and this is noted in the closing remarks as the referee suggests. On >the other hand, it is quite beyond the scope of the current paper to be presenting further details of these new >coordinated studies as they still have to be discussed and agreed within the QBOi community.

Page 18,: Lines 3-6: The importance of numerical advection schemes is noted but the specific schemes used are not given for each model. Also, could more detail be given

about the relation between the BD-Circulation and specific advection schemes? Do certain schemes consistently give a faster BD circulation?

>An additional table (Table 7) has been added to the revised manuscript giving the specific advection scheme used for >each model. The table also provides a brief summary of those artificial aspects of the model dynamical cores that >contribute to the dissipation. The subsection on advection has been rewritten and renamed "dynamical cores." A >reference for the sensitivity of the QBO to the BD-circulation is now given.

Page 19, Line 3: The title of the subsection includes the phase ". .. .feedbacks when interactive chemistry is included", however the subsection mostly discusses the ozone climatology for use without interactive chemistry. Maybe the title should be modified or more discussion of potential ozone feedback added.

>The title of this subsection has been changed to "Ozone."

Page 19, Section 5.1: this section discusses only a subset (13) of the QBOi models. While mentioned in Appendix B, maybe a sentence could be added here to note this restriction. In particular, are the results likely to be different for the Lindzen schemes? Also, I think this should be Section "5" rather than "5.1".

>It is now clear from the revised text that only a subset of QBOi models is discussed in subsection 5.1. It's only a >subsection so 5.1 is correct

Page 23, Line 3: "ENSO"; This might be a good place to reference a noted connection between the QBO disruption and ENSO: Barton, C. A., & McCormack, J. P. (2017). Origin of the 2016 QBO disruption and its relationship to extreme El Niño events. Geophysical Research Letters, 44. https://doi.org/10.1002/2017GL075576.

>Reference to work of Barton and McCormack is now included in the paper.

The following references in the reference section do not appear to be in the text: Page 30, Line 24: Giorgetta et al., 2013 Page 30, Line 35: Hazeleger et al., 2012 Page 34, Line 1: Stevens et al., 2013 Page 34, Line 21: Warner and McIntyre, 2001

>Hopefully all spurious references have now been removed from the references.

Page 30, Lines 15-19: The "a" and "b" are missing from the two Gerber et al. 2016references. Page 31, Line 24: The "a" is missing form the Kim et al. 2015 reference

>Corrected.

Page 33, Line 28: Serva et al., is misplaced in the reference list. >Moved.

---

## Author Comment (AC2) · 12 Jan 2018

Summary: The paper lies out the rationale for QBOi, defines several experiments that will allow scientific questions about the QBO to be answered, and gives some detailed information on the participating models. Not being an outright expert on the QBO, my view is that the experiments have been well thought out and that the paper will form an adequate basis for QBOi. Weaknesses of the paper include that in a few places unnecessary options are given to the participants that may complicate the evaluation of the results. Also some forcings could be specified in more detail, in order to reduce ambiguity. Furthermore, I am not sure enough relevant detail is conveyed about the

internal make-up of the models to allow analysts to understand why models behave the way they do. Maybe an additional appendix could be written, consisting of one-paragraph descriptions of the individual models, highlighting particular aspects of their make-ups that the PIs consider to be relevant for their simulation of the QBO, that go beyond just their representation of parameterized GWD. Finally, in some places I was unclear as to the function of this paper: The title suggests that the paper is about defining the experiments and describing the models, but in places the text talks about the performances of the models, which I consider out of scope for this paper. Such a discussion of model performance should be reserved for subsequent papers that conduct the scientific analysis.

>Again the authors thank the reviewer for their interesting comments. Unfortunately there appears to have been a slight misunderstanding. The protocol for phase 1 of QBOi has already been discussed and agreed by the community (see Hamilton et al., 2015 and Anstey et al. 2015) and in many cases the experiments have already been completed. Hence it is not appropriate to delete options from the phase 1 protocol (i.e., what the reviewer refers to as "unnecessary options" might actually be options that have been followed by some groups). The purpose of this manuscript is simply to document the protocol that is been used in phase 1 and give the scientific rationale used to agree that phase 1 protocol. Possible revisions and extensions of the protocol for the next phases of QBOi are already considered in the "closing remarks." Further it was a deliberate decision for QBOi to be less prescriptive than CMIP in order to encourage wider participation. Again this point is already made in the second paragraph of the Introduction, but we don't consider this paper to be the right place for a more detailed discussion of the relative merits of these different approaches. We agree that the QBOi approach does include ambiguities in some forcings (i.e., ozone) however this is mitigated for by the fact that the models are tuned to give their best QBO with that forcing. This point is now emphasised in the revisions to the subsection on ozone. The important thing from a QBOi perspective is that the same forcings (apart from the $CO_2$ amounts and SSTs), together with their concomitant tunings, are used across all

the experiments performed by each model (see page 5).

>We don't consider an additional one-paragraph description of each model will provide significantly more information than is already given in the tables, especially as an additional table has now been added on dynamical cores (Table 7 in the revised manuscript) . In our opinion these extra paragraphs would be just duplicating information that can be better obtained from the cited references for each model but with the risk that with the duplication possible mistakes and ambiguities are introduced.

>The function of the paper is quite clearly stated in the last three sentences of the abstract. While we agree that subsection 5.1 does diverge into a comparison of model performance (or at least parameterisation performance) and ideally merits a separate a paper the co-author who performed the comparison is no-longer in a position to prepare such a manuscript. As this basic comparison of the underpinning model features will be critical for the analysis papers we therefore consider it is essential that this material remains in this manuscript. Neither of the other 2 anonymous referees had any criticism of including GWD material.

P1L2: Capitalize "General Circulation and Earth System Models".

>Lower case is correct here but we are happy for the sub editor to make the change if that is considered appropriate for the journal's style.

P1L4: "Verisimilitude" is an unusual word in this context. While I appreciate stylistic richness, perhaps consider replacing with something more widely understood, such as "realism".

>The first draft of this paper had "realism" but the consensus among the lead authors was this wasn't strictly correct in this context and "verisimilitude" was considered more accurate. "Verisimilitude" is also preferred to all the alternative synonyms given in the Thesaurus we consulted.

P2L9: has -> have

>Unchanged

P2L24: has -> have

>Changed

Figure 1: Perhaps a sentence can be written about the Unified Model family.HadGEM2-CC produces a realistic QBO, HadGEM2-AO, HadGEM2-ES, and the AC-CESS models do not. What is it about HadGEM2-CC that produces this difference in behaviour? Higher lid, more levels in the stratosphere?

>It is quite beyond the scope of this paper to analyse the QBO or lack of QBO in the individual CMIP5 models or, indeed, model families (there are other families in addition to the Unified Model) shown in Figure 1. On the other hand we provide the more generic statement in Section 2, more appropriate in the multi-model context, that it is the introduction of GWD parameteriszation and increased resolution in the stratosphere that improves the ability of models to reproduce QBO-like tropical variability

P4L31: Cut out "fragile – which is to say"

>"Fragile" usefully emphasises the point that the ability of models to reproduce QBO-like behaviour is easily broken by a small change in formulating some aspects of models and therefore we have retained "fragile – which is to say."

P5L18: Insert "," after "experiments".

>Inserted.

P6L4: You say elsewhere that a 100-year simulation would be preferred. I suggest to remove the options of 3x30 years here and elsewhere. Otherwise you just complicate the analysis and might introduce problems with ensuring that the 3 ensemble members are sufficiently independent of each other.

> See comments above. The protocol has already been discussed and agreed by the community and it is not appropriate to change it at this stage in the project.

Figure 2: I suggest not to fill in the area under the curves in red, blue and black. These areas are overlapping, impossible to see in places and difficult to distinguish. One way to simplify this is to recognize that black is the mean annual cycle (i.e. it's periodic). If you subtract this off the other two signals and display the difference, the figure may become easier to read.

>Figure 2 has been redrawn in the rather different style suggested by Anonymous Referee # 3

P7L4: For the purpose of clarity, perhaps spell out explicitly what the 1xCO2, 2xCO2,and 4xCO2 mixing ratios actually are (e.g. 300 ppmv, 600 ppmv, 1200 ppmv, or whatever the numbers should be). Also what are the mixing ratios for other GHGs (CH4, N2O, etc.) for these experiments?

>Full details of the GHG gas forcings are given in Appendix A. For this part of paper we prefer to keep things simple and just refer to 1xCO2, 2xCO2, and 4xCO2, as it is difference between experiments that is the relevant information.

P7L19: Insert "some" before "models".

>Inserted.

P7L25: I would replace "9-12 months" with 12 months. Otherwise you may get an unnecessary diversity of responses there. Also spell out which reanalyses are supposed to be used here.

>As noted above the protocol is already agreed and includes the 9-12 months option. Full details of the experiment setup are given in Appendix A where it is spelt out which reanalyses are supposed to be used here. Our approach is to keep the main text of the paper as simple and readable as possible and direct the GMD reader to the Appendix and Supplemental for the details.

P8L14: ditto.

>Ditto.

P9L22: What is a "vertical resolution equivalent to the model resolution"? Would it not be easier to say that data are requested on model (hybrid-pressure, in most cases) levels? In this case surface pressure and a recipe for the calculation of model level pressure need to be provided, or full pressure fields for hybrid-height models.

>We have added "(i.e., with the same number of levels in the specified altitude range)" after "vertical resolution equivalent to the model resolution." As the review notes not all the models use hybrid-pressure vertical coordinates and we wanted to specify something that would apply to all the models.

P12L6: Experiment 5A is designed for coupled AOGCMs, so the statement is inaccurate, but maybe models without an option to couple to an ocean can just skip this experiment. (Experiment 5A talks about an "appropriate initialization" of the ocean: What is that? Please provide more detail.)

>The statement is in fact correct. ALL the experiments have been designed for atmosphere-only models. However what we say is that one of the experiments (Experiment 5) can also be performed with a coupled model, in which case the experiment has been renamed Experiment 5a. In the spirit of QBOi the aim is to encourage as wide as possible participation hence the project will also consider results from coupled models for Experiment 5. "Appropriate initialise" means the ocean should also be initialised but by whatever method is considered appropriate for that particular model. Again QBOi aims not to over prescribe the experiments.

P19L3: I do not understand why the ozone forcing for models without interactive chemistry is not prescribed. This will be an uncontrolled and unnecessary source of inter-model variations, compounded by the lack of any requirements to document any ozone forcing used. Since almost all models listed will not have interactive ozone, prescribing ozone, and perhaps also conducting a sensitivity experiment using a variant ozone climatology, might be interesting to include. It would be straightforward e.g. to require

models without interactive ozone chemistry to use the CMIP6 historical ozone climatology. An option here is to request variant simulation where ozone is varied from CMIP6 to whatever individual groups prefer, to get an idea about the influence of this choice.

>The subsection on ozone has been rewritten but we emphasise that QBOi has deliberately chosen not to be over prescriptive. For phase 1 of the QBOi analysis this sensitivity to precise details of the ozone climatology was not considered important since all the models are tuned to give their best QBO. The importance was in ensuring that each model used the same ozone across all the experiments performed, as is already emphasised in Section 3. Sensitivity to ozone will be considered in the next phases of the QBOi as noted in the closing remarks of the paper.

P24L12: Again, I suggest to replace the two options with a simple requirement to produce a single 100-year simulation.

>Again we note that the experimental protocol has already been agreed and it is not appropriate to remove one of the options.

P24L23: To remove ambiguity and make everyone's lives easier, the CO2 volume mixing ratio reached in 2002 should be numerically stated here. (How about the other GHGs?)

>If the model includes other GHGs it is already stated a few lines earlier that the 2002 CMIP5 forcings should be used.

P25L15: Replace "9-12 months" with "12 months". >9-12 months is in the agreed protocol so no change has been made.

P26L9: Why not require the data on ECMWF model levels, and ditto for other fields? This way no information is lost to interpolation.

>Although input winds and temperatures on ECMWF model levels could have been used, it was an unnecessary complication since any information lost in interpolation is not important as the profiles are simply meant to be representative of typical observed

profiles.

Figure 7, section 5.1, appendix B: This is moving into the territory of comparing model performance, which I think would be out of scope for this paper. This off-line comparison is actually an interesting scientific exercise itself, but does not quite fit into this paper, the thrust of which is on describing the QBOi experiments. The brevity of presentation afforded to it here also does not quite do it justice. Please consider expanding this into a compact stand-alone publication (which might be submitted to GMD and would be an interesting companion paper to this one).

>We agree a separate paper on the GWD might be valuable but the reasons for not doing this are given in the general comments at the beginning of this response. Also it is not really in the territory of comparing model performance but more in the territory of comparing how GWD is represented in models in the same way as we compare how resolution is represented in the models.

---

## Author Comment (AC3) · 12 Jan 2018

This study introduces the model integrations performed as part of the first phase of the QBOi, a model inter comparison project that hopes to shed light on the processes that lead to a spontaneous QBO and how they will change in the future. Such a project is sorely needed in our field, and I look forward to reading future papers that utilize this model output. I have only a few minor comments, and after the authors address them the paper will be ready for publication.

>Again the authors thank the referee for kindly reviewing this manuscript.

[Figure]

1. Figure 2: The use of filled black patches for the "mean annual cycle" forcing is visually confusing. I suggest a thick line.

>The suggested change has been made.

2. It is too late to correct this, but in retrospect there probably should have been guidance for the ozone profile to be used for models without interactive chemistry. There are ozone-temperature feedbacks in this region that will differ among models, and unraveling the causes of these feedbacks will likely be hard. Again, I don't think it is worth rerunning experiments, and hopefully the archived ozone will suffice.

>The experimental protocol has already been agreed and can not be changed for phase 1 of QBOi, however in the closing remarks we allow for the possibility of testing the QBO's sensitivity to ozone in the next phases of QBOi. As noted also in our response to reviewer #2, in phase 1 of QBOi the models are tuned to give the "best" QBO and sensitivity to the precise details of the ozone profile was not considered important provided that, for each model, the same ozone profile was used across all experiments. This is now spelt out more clearly in the revised subsection on ozone.

3. The numerical, thermal, and mechanical dissipation used by each model likely differs, and these three sources of dissipation might be important for the QBO momentum budget in some models (e.g. Yao and Jablonowski,2015, already cited). I have two suggestions: first, please ask the models to submit their wind and temperature tendencies due to these three sources of dissipation (or at least the total tendency due to dissipation)! Zonal and monthly mean is probably good enough. Second, please add a column to table 6 or 7 (or a new table) where each model reports on how it implements numerical, thermal, and mechanical dissipation. It might also be helpful for each model to state which advection scheme/dynamical core it uses.

>We have added a new table (Table 7 in the new manuscript) in which each model reports its advection scheme and artificial dissipation processes. Again the data request has already been agreed and groups have mostly finished uploading their output.

[Figure]

Nonetheless we note and agree with the referee's suggestion for archiving tendency terms and will try to include these in all future QBOi data requests.

4. Figure 7, top left panel: I suggest writing the model names in color. Also, two colors appear to be used for more than one model (at least to this reviewer's mildly color-blind eye). Specifically, the shade of red used for MIROC-ESM (F-H) and HadGEM2-AC (P-WM) is very similar. Similarly, the shade of green used for LMDz6 (P-L) and EMAC (F-H) is very similar. Please adjust the colors to add more contrast.

>For consistency with the other figures, we have not written the model names in color, however labelling has been improved in this figure (and Figures 4 & 6) by increasing the size of the line segments in the legend. The grey of the wind profile has also been lightened so it doesn't conflict with the legend text as harshly. Model color allocation has been agreed for all the phase 1 QBOi papers and therefore changing it in this figure is perhaps not a good idea. It is rather difficult finding 17 distinct colors and we accept that this sometimes makes it difficult to distinguish models but on our careful checking it is always possible for the figures in this paper.

5. Page 23, line 4 "between" is misspelled.

>Corrected.

6. Table 4: it would be helpful if one level near 200hPa was also included, as one might want to compare the upper tropospheric resolved wave spectrum (i.e. Wheeler and Kiladis 1999 diagrams) among models. That is, the resolved wave spectrum near the top of convection will differ among the models (possibly due to different convection schemes used by each model), and it might be interesting to relate any differences in QBO morphology to differences in tropospheric wave generation that are in turn related to convection schemes. This additional level will also allow one to study the affect of vertical resolution in the TTL on resolved wave vertical propagation – it is conceivable that models with coarser vertical resolution will have stronger degradation in their resolved wave fluxes between âĹij200hPa and âĹij100hPa.

>Again the data request has already been agreed and groups have uploaded data. Nonetheless we agree with the reviewer's suggestion and will consider adding upper tropospheric levels to future QBOi data requests.

7. The native vertical levels of each model (i.e. the data underlying figure 4) should be made accessible, perhaps as a data supplement or hosted on the QBOi website.

>The location of the native vertical levels of each model is now in indicated in Figure 4b by the very thin horizontal lines.